# DecompDreamer: A Composition-Aware Curriculum for Structured 3D Asset Generation

**Utkarsh Nath**[*][†]                                                                                     *unath@linkedin.com*
*LinkedIn*
*Arizona State University*

**Rajeev Goel**[*]                                                                                          *rgoel15@asu.edu*
*Arizona State University*

**Rahul Khurana**                                                                                          *rkhura11@asu.edu*
*Arizona State University*

**Kyle Min**                                                                                                *kyle.min@oracle.com*
*Oracle*

**Mark Ollila**                                                                                             *mark.ollila@asu.edu*
*Arizona State University*

**Pavan Turaga**                                                                                            *pturaga@asu.edu*
*Arizona State University*

**Varun Jampani**                                                                                          *varunjampani@gmail.com*
*Arcade AI*

**Tejaswi Gowda**                                                                                          *tejaswi@asu.edu*
*Arizona State University*

**Reviewed on OpenReview:** *https://openreview.net/forum?id=3qy4J6QFbn*

## Abstract

Current text-to-3D methods excel at generating single objects but falter on compositional prompts. We argue this failure is fundamental to their optimization schedules, as simultaneous or iterative heuristics predictably collapse under a combinatorial explosion of conflicting gradients, leading to entangled geometry or catastrophic divergence. In this paper, we reframe the core challenge of compositional generation as one of optimization scheduling. We introduce DecompDreamer, a framework built on a novel staged optimization strategy that functions as an implicit curriculum. Our method first establishes a coherent structural scaffold by prioritizing inter-object relationships before shifting to the high-fidelity refinement of individual components. This temporal decoupling of competing objectives provides a robust solution to gradient conflict. Qualitative and quantitative evaluations on diverse compositional prompts demonstrate that DecompDreamer outperforms state-of-the-art methods in fidelity, disentanglement, and spatial coherence. Code available at:
https://github.com/rajeevgl01/decompdreamer

---

[*]Equal contribution. Correspondence to rgoel15@asu.edu
[†]Work done while at Arizona State University

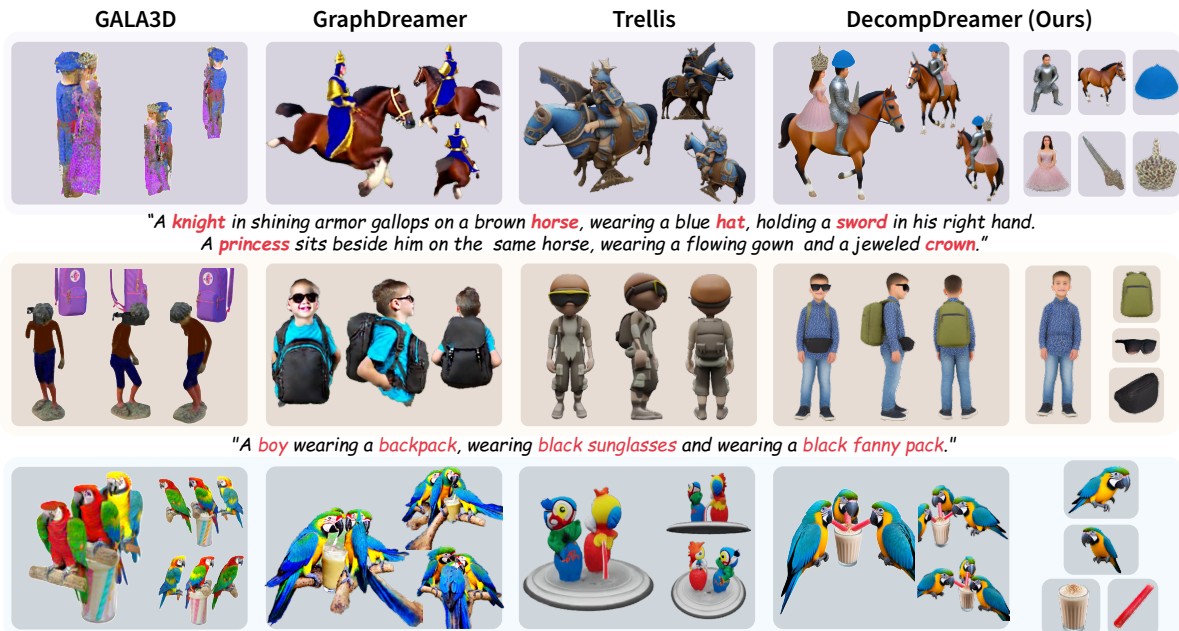

Figure 1: Illustration of high-quality compositional 3D assets generated by DecompDreamer for complex text prompts. Existing methods often miss objects or distort spatial relationships, while DecompDreamer accurately captures geometry and preserves inter-object layout.

# 1 Introduction

The generation of complex, multi-object 3D scenes from natural language Bai et al. (2023); Gao et al. (2024); Ge et al. (2025); Po & Wetzstein (2024), or compositional text-to-3D, is fundamentally a high-dimensional combinatorial optimization problem. The task requires discovering a single plausible configuration from a virtually infinite search space of object geometries, attributes, and spatial relationships, guided only by the sparse constraints of a text prompt. The complexity of this search space grows exponentially with the number of objects and relationships, rendering naïve optimization strategies intractable. Common failure modes, such as misattributing colors to the wrong objects or generating physically implausible spatial arrangements, are the predictable outcomes of navigating this extraordinarily complex and ill-posed optimization landscape.

To make this problem tractable, existing methods Ge et al. (2025); Gao et al. (2024) rely on various optimization heuristics; however, they are often limited by a shared underlying pathology: conflicting gradients Yu et al. (2020). Compositional generation can be viewed as an implicit multi-task learning Caruana (1997); Kendall et al. (2018) problem where each object and relationship constitutes a distinct "task". When multiple objectives, such as generating a high-fidelity object, binding its attributes correctly, and positioning it relative to others, are pursued simultaneously, the resulting supervisory signals can interfere destructively. Holistic methods Poole et al. (2023); Liang et al. (2024); Xiang et al. (2025) that treat the scene as a single entity suffer from severe gradient conflicts between implicit object tasks. More advanced methods Gao et al. (2024); Zhou et al. (2024) that decompose the scene using spatial layouts or semantic scene graphs mitigate this to some degree, but their reliance on simultaneous or joint optimization schedules re-introduces the problem. They attempt to solve for global structure and local detail concurrently, leading to an intractable multi-task problem that struggles to scale and often results in entangled or blended geometry.

This analysis reveals that the critical, unaddressed bottleneck in compositional generation is not just the static decomposition of the scene, but the dynamic scheduling of the optimization process. In this paper, we propose DecompDreamer, a framework that introduces a staged, composition-aware optimization strategy that resolves this fundamental challenge. Our approach is framed as an implicit curriculum Bengio et al. (2009) that transforms the intractable problem of joint generation into a sequence of manageable sub-problems.

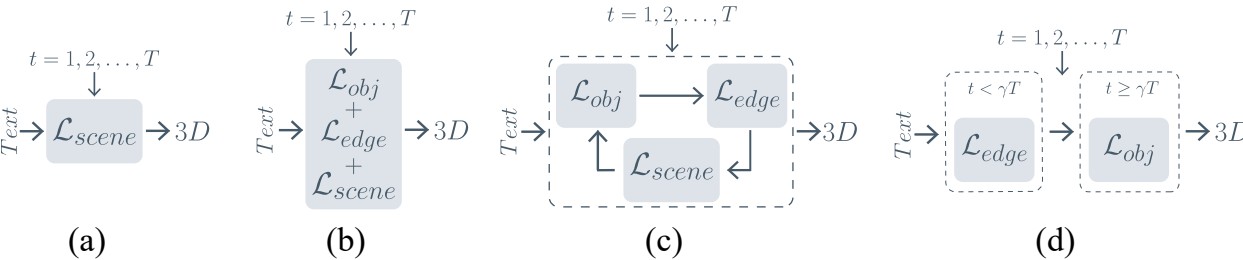

Figure 2: A visual taxonomy of optimization heuristics. (a) Holistic uses a single global loss. (b) Simultaneous applies all losses concurrently. (c) Iterative applies losses in a sequential loop. (d) Our staged curriculum temporally decouples relational and object losses

Inspired by the traditional 3D art workflows, we establish a scene's global structure before refining individual details Chopine (2011), our method operationalizes this into a two-stage optimization schedule. In Stage One: Joint Relationship Modeling, we first prioritize the establishment of a coherent structural scaffold by focusing the optimization on inter-object relationships. This solves for the low-frequency components of the scene, effectively constraining the search space for the next stage. In Stage Two: Targeted and Disentangled Refinement, with the global structure fixed, we shift focus to refining the high-frequency details of individual objects. This stage employs targeted relational losses and view-aware supervision to enhance fidelity and enforce disentanglement without disrupting the previously established coherence.

By temporally decoupling the optimization of competing objectives, our staged approach provides a robust, structural mechanism for mitigating gradient conflict. This allows DecompDreamer to generate high-fidelity, well-disentangled 3D assets from highly complex prompts involving numerous objects and intricate interactions.

Our contributions can be summarized as:

- We introduce a novel staged optimization strategy, framed as an implicit curriculum, that transforms the intractable problem of joint compositional generation into a sequence of manageable sub-problems. This "Composition-Aware" schedule first establishes global relational structure and then refines local object detail, providing a robust solution to the gradient conflicts that limit prior work.

- We propose a set of techniques for the detail-refinement stage, including targeted relational optimization and view-aware supervision, that achieve strong object disentanglement by preventing feature blending and ensuring geometric consistency without disrupting the globally coherent structure established in the first stage.

- We demonstrate through extensive qualitative and quantitative comparisons that our staged optimization paradigm significantly outperforms state-of-the-art methods in fidelity, coherence, and scalability, particularly on complex prompts with numerous objects and intricate interactions.

## 2 Theoretical Foundations

### 2.1 The Combinatorial Complexity of the Task

Compositional text-to-3D generation is a high-dimensional combinatorial optimization problem Du et al. (2020): a search for an optimal scene configuration $S$ that maximizes a plausibility function $P(S|T)$ given a text prompt $T$. The configuration $S$ is a complex tuple of objects $\{O_i\}$, attribute bindings $A : \{O_i \to \{a_{ij}\}\}$, and relational predicates $R : \{(O_i, O_j) \to \{r_k\}\}$. The search space for $S$ is combinatorially vast. For a prompt like *"A blue bird sitting on a black branch of a white tree,"* the model must solve the discrete assignment problem of binding attributes to entities. An incorrect binding can still yield a plausible rendering from certain views, creating a poor local minimum. Furthermore, satisfying relational predicates like "sitting on" imposes strong geometric constraints on the joint parameter space of the objects. The number of such

interdependent variables grows exponentially, making the problem intractable without effective heuristics to structure and prune the search.

## 2.2 A Taxonomy of Optimization Heuristics

The intractability of this problem necessitates the use of optimization heuristics. We propose a taxonomy of prior work based on how each strategy applies object-level ($\mathcal{L}_{\text{obj}}$), relational ($\mathcal{L}_{\text{edge}}$), and scene-level ($\mathcal{L}_{\text{scene}}$) objectives.

**Heuristic 1: Holistic Optimization.** This baseline approach Xiang et al. (2025); Liang et al. (2024); Nath et al. (2025); Lin et al. (2023); Shi et al. (2024); Zhu et al. (2024); Chen et al. (2023) treats the scene as a monolithic entity, driven by a single objective, $\mathcal{L}_{\text{total}} = \mathcal{L}_{\text{scene}}$. It imposes no explicit compositional structure and is highly susceptible to conflicting gradients from the multiple implicit tasks, leading to entangled representations and incorrect attribute binding.

**Heuristic 2: Simultaneous Decomposed Optimization.** This heuristic decomposes the scene but optimizes all components concurrently via a weighted sum: $\mathcal{L}_{\text{total}} = w_1\mathcal{L}_{\text{obj}} + w_2\mathcal{L}_{\text{edge}} + w_3\mathcal{L}_{\text{scene}}$. Methods like Gala3D use strong structural priors (e.g., 3D bounding boxes) that prevent catastrophic failure. However, the simultaneous optimization of object geometry and scene layout, combined with the rigidity of the priors lead to a predictable limitation: stable but suboptimal convergence. This is particularly evident in complex interactions that cannot be fully captured within simple spatial containers, such as overlapping or interdependent object relationships (e.g., *"knight riding horse"*).

**Heuristic 3: Iterative Decomposed Optimization.** This approach also decomposes the scene but applies objectives in an alternating loop within each iteration (e.g., (i) $\mathcal{L}_{\text{obj}}$, then (ii) $\mathcal{L}_{\text{edge}}$, etc.). This heuristic, used by methods like GraphDreamer Gao et al. (2024), DreamHOI Zhu et al. (2025) and CompGS Ge et al. (2025), is vulnerable to optimization divergence at scale. As scene complexity grows, the combinatorial explosion of conflicting gradients creates a chaotic "tug-of-war," causing the averaged gradient to become nonsensical. This leads the process to diverge rather than converge, resulting in a total failure of generation.

**Heuristic 4: Staged Decomposed Optimization (DecompDreamer).** This paradigm, which we introduce, is a structured, multi-stage curriculum Bengio et al. (2009); Caruana (1997); Kendall et al. (2018) that avoids optimizing the entire scene at once. Instead, it first tackles coherent sub-problems by prioritizing relational objectives ($\mathcal{L}_{\text{joint}}$) to establish a global scaffold (Stage 1: Structure-focused). Only then does it shift focus to refining individual components with object-level and targeted losses ($\mathcal{L}_{\text{obj}}, \mathcal{L}_{\text{target}}$) that preserve this structure (Stage 2: Detail-focused). This "structure-then-detail" approach ensures that the gradients within each stage are more self-consistent and minimally conflicting. By temporally decoupling competing objectives, it avoids the optimization divergence plaguing iterative methods, ensuring stable and scalable convergence. Our ablation study (Figures 5, 7) provides empirical validation, showing a non-staged variant fails to produce structurally consistent results.

# 3 Related Work

## 3.1 Text-to-3D Generation

Methods for generating 3D content from text descriptions are broadly categorized into two paradigms: 3D supervised (feed-forward) Cheng et al. (2023); Jun & Nichol (2023); Nichol et al. (2022); Wei et al. (2023); Xiang et al. (2025)techniques and 2D lifting (optimization-based) techniques Chen et al. (2024; 2023); Huang et al. (2024); Poole et al. (2023); Wang et al. (2023); Zhu et al. (2024). 3D supervised methods train models directly on paired text and 3D data, enabling the generation of high-quality 3D assets. However, the efficacy of these approaches is fundamentally limited by the availability and diversity of large-scale 3D datasets Deitke et al. (2023). As most existing datasets consist primarily of single-object models, these methods struggle to generalize to complex, compositional scenes involving multiple objects and interactions. 2D lifting techniques circumvent the need for large 3D datasets by instead leveraging powerful, pre-trained 2D diffusion models to guide the optimization of a 3D representation Mildenhall et al. (2021); Kerbl et al. (2023). The seminal work in this area, DreamFusion Poole et al. (2023), introduced Score Distillation Sampling (SDS), a method for

aligning rendered views of a 3D model with a 2D diffusion prior. This optimization-based paradigm inspired numerous advancements Liang et al. (2024); Lin et al. (2023); Tang et al. (2024; 2023); Wang et al. (2023). However, these foundational methods typically treat the entire scene as a single, monolithic entity. While effective for individual objects, they lack explicit mechanisms for modeling inter-object relationships and thus underperform significantly on compositional prompts.

## 3.2 Compositional 3D Generation

To address the limitations of monolithic generation, a dedicated subfield Dhamo et al. (2021); Bai et al. (2023); Po & Wetzstein (2024); Zhai et al. (2023) has emerged focusing on compositional scenes, which can be analyzed using the framework we introduced in Section 2.2. An initial wave of research exemplifies what we classify as Simultaneous Decomposed Optimization (Heuristic 2). These methods integrate explicit layout priors, such as non-overlapping 3D bounding boxes, to provide object-level guidance. Gala3D Zhou et al. (2024), for instance, is a canonical example that adopts a layout-guided approach. While these methods effectively prevent catastrophic failures, their reliance on rigid spatial representations often results in stable but suboptimal convergence. Empirical results indicate that they struggle to capture complex interactions where object geometries must overlap plausibly (e.g., "a knight riding a horse"). Subsequent work moved towards more flexible semantic decomposition strategies, often employing Iterative Decomposed Optimization (Heuristic 3). GraphDreamer was a key advance, proposing the use of a scene graph to model objects and their relationships separately. However, its iterative optimization schedule is susceptible to optimization divergence: as the number of objects increases, the combinatorial explosion of conflicting gradients can lead to catastrophic failures. This is consistent with empirical observations, where GraphDreamer often fails on prompts containing more than four objects, producing incoherent or nonsensical outputs. Our work, DecompDreamer, introduces a novel approach with Staged Decomposed Optimization (Heuristic 4). Like recent methods Gao et al. (2024); Zhou et al. (2024), we use 3D Gaussians and a semantic scene graph. However, we address the optimization-based failures of prior heuristics by introducing a principled curriculum that temporally decouples competing objectives. This staged approach, which shifts from modeling inter-object relationships to refining disentangled objects, is designed to ensure stable and scalable convergence where other heuristics fail.

**Temporal Decoupling in Generation.** Recent works have highlighted the importance of temporal scheduling in generative models. eDiff-I Balaji et al. (2022) introduced an ensemble of expert denoisers specialized for different synthesis stages, while ThemeStation Wang et al. (2024) utilized temporally decoupled score distillation to maintain thematic coherence. These approaches validate our hypothesis that the optimization of global structure must be decoupled from local details. DecompDreamer operationalizes this insight to resolve the combinatorial gradient conflicts unique to multi-object compositional scenes, a challenge not addressed by prior heuristics.

## 4 Preliminaries

**Gaussian Splatting** Kerbl et al. (2023) represents a scene as a set of 3D Gaussians defined by means, covariances, colors, and opacities. These are projected as 2D Gaussians and composited via alpha blending in front-to-back order. Compared to NeRF-based methods Barron et al. (2022); Müller et al. (2022), this approach offers higher fidelity, faster rendering, and lower memory usage. The rendering process is defined as $r = g(\theta, c)$, where $\theta$ is the set of Gaussians, $g$ is the renderer, and $c$ the camera parameters.

**Score Distillation Sampling (SDS)** Poole et al. (2023) optimizes a 3D representation $\theta$ using guidance from a pre-trained 2D diffusion model $\phi$. At each iteration, an image $z$ is rendered from $\theta$ using a random camera viewpoint and a view-dependent text prompt $y_\psi$, where $\psi$ denotes the sampled azimuth angle. The diffusion model provides a supervisory gradient to update $\theta$ such that its renderings align with the prompt. The optimization is guided by the SDS gradient:

$$\nabla_\theta \mathcal{L}_{\text{SDS}}(z; y_\psi) = \mathbb{E}_{t,\epsilon} \left[ \omega(t) \left( \hat{\epsilon}_\phi(z_t, t, y_\psi) - \epsilon \right) \frac{\partial z}{\partial \theta} \right], \tag{1}$$

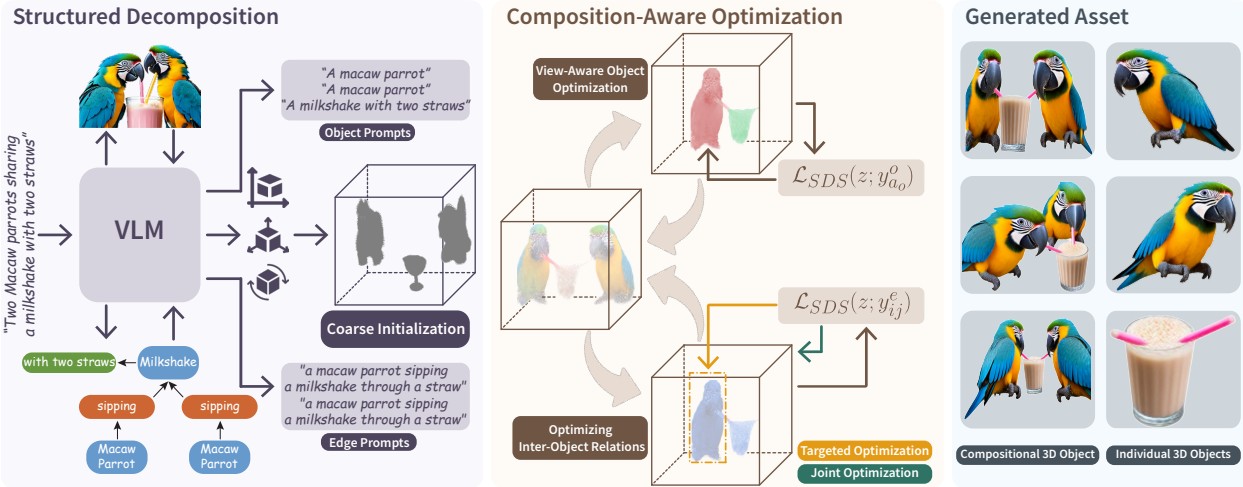

Figure 3: Overview of the DecompDreamer pipeline. Given a text prompt, a VLM generates a scene graph to guide a coarse initialization. The core of our method is a composition-aware optimization curriculum that first models joint relationships to build a coherent structure, then refines individual objects to produce high-fidelity, disentangled 3D assets.

where $\hat{\epsilon}_\phi$ is the noise predicted by the diffusion model, $z_t$ is the noisy version of the rendered image, and $\omega(t)$ is a weighting function. This process distills the knowledge of the 2D model into the 3D representation. In our work, we replace the standard SDS loss with supervision from a flow model AI (2023), which learns a direct vector field from noisy to clean images for improved gradient stability.

# 5 Method

Our objective is to generate high-quality 3D assets for complex text prompts that describe intricate inter-object relationships. Figure 3 shows an overview. Section 5.1 details our structured decomposition using a VLM. Section 5.2 describes the full Composition-Aware optimization pipeline for 3D asset generation.

## 5.1 Structured Decomposition with VLMs

Before designing a complex 3D asset, artists typically plan the layout, size, and orientation of objects. Inspired by this workflow, we use a Vision-Language Model (VLM), specifically ChatGPT-4o OpenAI (2024), to construct a compositional scene graph guiding 3D generation. Given a text prompt $t^g$, the VLM produces an image $I$ and a scene graph $G = (V, E)$, where $V = \{v_i\}_{i=1}^n$ represents objects $\{O_i\}_{i=1}^n$ and $E$ encodes inter-object relationships. The image $I$ provides a visual blueprint to initialize object-level spatial attributes. Each object $O_i$ is associated with attributes $\{a_{ij}\}_{j=1}^m$ to construct an object prompt $t_i^o$, Similarly, relationships between object pairs $(O_i, O_j)$ are converted into *edge prompts* $t_{ij}^e$, composed from the object names and their described interaction. Using $I$ as reference, we estimate object center coordinates, size, orientation, and azimuth angles, which are used in the View-Aware Disentangled Object Optimization module to supervise individual objects and ensure consistent, disentangled appearance.

**Coarse Initialization.** Each object $O_i$ is represented by a distinct 3D Gaussian cloud, initialized from TripoSR Tochilkin et al. (2024)) point clouds and aligned using VLM-inferred spatial properties. For efficient disentangled optimization, we maintain a mapping that tracks Gaussians corresponding to each object, enabling independent refinement of properties such as mean, covariance, and opacity. Further implementation details are in Section A.4.

## 5.2 Composition-Aware Optimization

We begin by optimizing objects with respect to inter-object relationships before gradually shifting to targeted object refinement. Section 5.2.1 details two strategies for modeling inter-object interactions, while Section 5.2.2 addresses view-aware object modeling for disentanglement. Finally, Section 5.2.3 outlines the full optimization pipeline, integrating relationship modeling and object refinement for a coherent 3D representation.

### 5.2.1 Ensuring Coherent Inter-Object Interactions

In our framework, inter-object relationships are represented using edge prompts, which describe the contextual interactions between object pairs. To effectively model these relationships, we employ two complementary strategies for optimizing the associated Gaussians:

**Joint Optimization of Related Objects:** Given an edge prompt $t_{ij}^e$ describing the relationship between objects $O_i$ and $O_j$, we jointly optimize the Gaussians corresponding to both objects. This ensures that the spatial and semantic relationships between the objects are preserved in the 3D scene. The joint optimization loss is the SDS loss applied to a rendered view containing both objects:

$$\mathcal{L}_{joint}(O_i, O_j, y_{ij}^e) = \nabla_{\mathcal{G}_{o_i}, \mathcal{G}_{o_j}} \mathcal{L}_{\text{SDS}}(z; y_{ij}^e), \tag{2}$$

where $z$ is the rendered image containing both objects, and $y_{ij}^e$ is the text embedding of the edge prompt $t_{ij}^e$.

**Targeted Optimization for a Single Object:** While joint optimization ensures cohesive inter-object relationships, it can sometimes result in undesirable blending, where parts of one object flow into another. To mitigate this, we employ targeted optimization based on the edge prompt, focusing the optimization solely on one object while keeping the other fixed. The goal is to adjust the selected object such that it adheres to the overall relationship without affecting the structural integrity of the other object. The SDS loss for targeted optimization is formulated as:

$$\mathcal{L}_{target}(O_i, y_{ij}^e) = \nabla_{\mathcal{G}_{o_i}} \mathcal{L}_{\text{SDS}}(z; y_{ij}^e), \tag{3}$$

where, only the Gaussians of object $O_i$ are optimized based on the edge prompt $t^{e_{ij}}$.

### 5.2.2 View-Aware Disentangled Object Optimization

DecompDreamer promotes object disentanglement by assigning each Gaussian to a single object, ensuring clear boundaries. However, optimizing only through inter-object relationships can lead to blending and suboptimal refinement of individual components. This section addresses how we optimize object attributes while preserving disentanglement using view-specific and negative prompts.

**Optimizing Objects in Compositional Settings.** In single-object generation, view-dependent text embeddings Poole et al. (2023) have proven effective for aligning 3D renders with orientation-specific details. However, in compositional settings, a single global camera view often fails to represent all components accurately—leading to conflicting gradients and misalignment between prompts and rendered views. To address this, we propose object-view-dependent supervision. For each object $O_i$, we estimate its angular offset $\phi_i$ from a canonical front-facing orientation (0° azimuth) using the VLM (workflow detailed in Appendix A.11). The corrected azimuth $\psi_i = \psi - \phi_i$ is used to compute the view-aligned embedding $y_{\psi_i}^o$. We incorporate this adjustment into our SDS loss, yielding the object-level optimization:

$$\mathcal{L}_{\text{obj}}(O_i, \ y_{\psi_i}^o) = \nabla_{\mathcal{G}_{o_i}} \mathcal{L}_{\text{SDS}}(z; \ y_{\psi_i}^o) \tag{4}$$

This refinement ensures that each object is supervised using a view-aligned prompt, improving consistency, orientation accuracy, and geometric coherence in the compositional output.

**Negative Prompt for Disentangled Object.** To prevent feature blending between related components, we introduce negative prompts during object-specific optimization. For a given object $O_i$, the negative prompt is constructed by aggregating the labels of all directly connected objects in the scene graph. This encourages the model to explicitly suppress features from neighboring components, thereby reinforcing object-level separation and preserving clean boundaries.

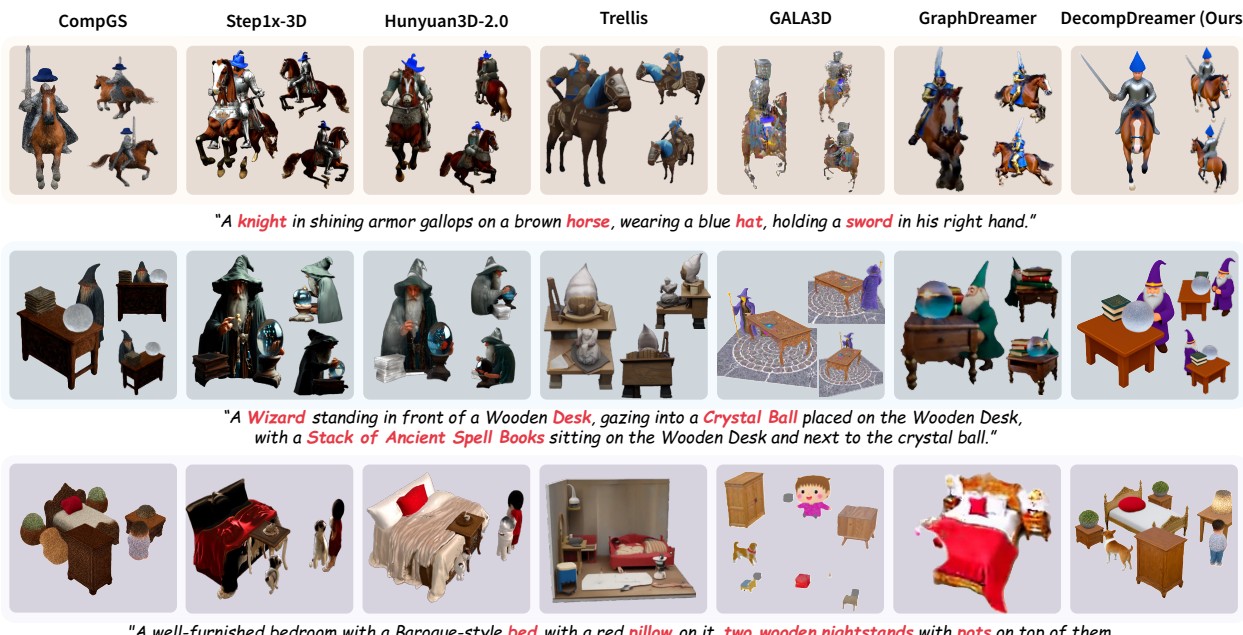

| CompGS | Step1x-3D | Hunyuan3D-2.0 | Trellis | GALA3D | GraphDreamer | DecompDreamer (Ours) |

*"A **knight** in shining armor gallops on a brown **horse**, wearing a blue **hat**, holding a **sword** in his right hand."*

*"A **Wizard** standing in front of a Wooden **Desk**, gazing into a **Crystal Ball** placed on the Wooden Desk, with a **Stack of Ancient Spell Books** sitting on the Wooden Desk and next to the crystal ball."*

*"A well-furnished bedroom with a Baroque-style **bed** with a red **pillow** on it, **two wooden nightstands** with **pots** on top of them, a square wooden **table** holding a decorative **table lamp**, a wooden **wardrobe**, a **kid** standing in front of bed, and a pet **dog** standing next to the bed between nightstand and wardrobe."*

Figure 4: Qualitative comparison between the proposed DecompDreamer and state-of-the-art text-to-3D generators. More results can be found in Section A.12 of the appendix.

### 5.2.3 Overall Optimization

**Spatial Error Correction.** Before optimizing the full compositional object, we correct spatial misalignments arising from VLM-predicted object centers. To this end, we introduce object-specific translation parameters $T_i^o = (x_i, y_i, z_i)$ for each object $O_i$. These parameters are applied to shift the corresponding Gaussians as $\mathcal{G}_{o_i} \leftarrow \mathcal{G}_{o_i} + T_i^o$. To isolate translation refinement, we optimize $T_i^o$ by minimizing joint edge-level losses:

$$\mathcal{L}_{\text{spatial-error-correction}} = \sum_{(i,j)\in E} \mathcal{L}_{\text{joint}}(O_i, O_j, y_{ij}^e) \tag{5}$$

while keeping all other parameters of $O_i$ and $O_j$ frozen. This adjustment ensures that relationship optimization operates on correctly aligned components, improving the geometric fidelity of the overall composition. After error correction, we begin by modeling inter-object relationships before gradually shifting focus toward individual objects. The optimization is divided into two stages:

**Stage One**: We emphasize joint modeling of inter-object relationships. For each edge in the scene graph, we apply joint optimization equation 2, while progressively increasing the weight of individual object optimization equation 4.

**Stage Two**: The focus shifts toward refining individual objects while maintaining inter-object consistency. Here, we perform targeted optimization for each object equation 3 along with individual object optimization equation 4. This transition allows fine-grained details to be added while preserving relationships established earlier. The effective optimization is:

$$\mathcal{L}_{\text{loss}} = \begin{cases} \mathcal{L}_{\text{joint}}(O_i, O_j, y_{ij}^e) + \lambda \left(\frac{t}{T}\right)^2 \sum\limits_{O\in\{O_i,O_j\}} \mathcal{L}_{\text{obj}}(O, y_{a_o}^o), & \text{if } t < \gamma T \\ \sum\limits_{O\in\{O_i,O_j\}} \left(\mathcal{L}_{\text{target}}(O, y_{ij}^e) + \mathcal{L}_{\text{obj}}(O, y_{a_o}^o)\right), & \text{if } t \geq \gamma T \end{cases} \tag{6}$$

where $(i,j) \in E$, $t$ represents the current iteration, $T$ is the total number of iterations, and $\gamma$ controls the transition point between joint and targeted optimization. The full pipeline, including algorithmic details, is provided in Section A.2 of the Appendix.

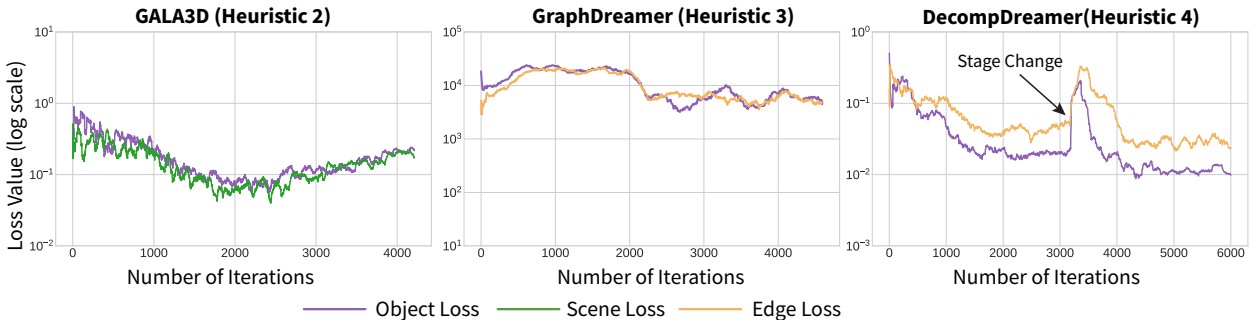

Figure 5: Empirical validation of optimization heuristics. The plots show the average object, scene, and edge losses for a complex 4-object text prompt ("A knight in shining armor..."). (a) GALA3D converges suboptimally, (b) GraphDreamer diverges, while (c) our staged approach (DecompDreamer) converges stably.

# 6    Experiments

We implement our framework in PyTorch, using a rectified-flow model (*stabilityai/stable-diffusion-3.5-medium*) for guidance and representing scenes with 3D Gaussian Splatting. Key hyperparameters, including guidance scale, number of Gaussians, and optimization iterations, are provided in Appendix A.3. Our experiments validate our approach through an analysis of loss dynamics (Section 6.1), comparisons with state-of-the-art methods (Section 6.2), and ablation studies (Section 6.3). A typical two-object scene is generated in 12 minutes on a single NVIDIA A100 GPU when initialized with TripoSR. A full runtime comparison with baselines is provided in Appendix 6.4, and we further discuss the effect of different initializers in Section 6.6.

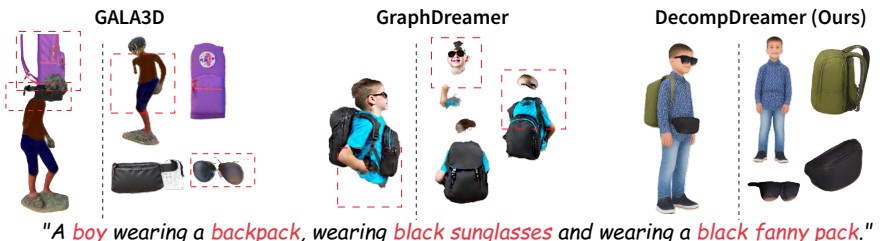

*"A boy wearing a backpack, wearing black sunglasses and wearing a black fanny pack."*

Figure 6: Qualitative comparison of object disentanglement in GALA3D, GraphDreamer, and DecompDreamer. Each method displays the full 3D object (left) and its components (right).

## 6.1    Quantifying Gradient Conflicts

To empirically validate our taxonomy, we analyze the loss dynamics of representative methods in Figure 5 and Figure 10 of the appendix. The results visually confirm the predicted failure modes of prior heuristics and demonstrate the stability of our staged approach. The loss curves for GALA3D (Figure 5a) illustrate the predicted stable but suboptimal convergence. After an initial descent, both object and scene losses hit a floor and begin to rise again, eventually settling at a high error value. This trajectory reveals the inherent conflict between optimizing object fidelity and adhering to rigid layout priors, leading the system to converge to a poor local minimum. The trajectory for GraphDreamer (Figure 5b) provides a clear illustration of optimization divergence. The object and edge losses rapidly increase, indicating the optimizer is immediately overwhelmed by a combinatorial explosion of conflicting gradients. This creates a chaotic "tug-of-war," visually evident as the loss curves cross while trending upwards, which suggests that an improvement in one objective comes at the direct expense of the other, leading to a failed optimization. In stark contrast, the loss curves for DecompDreamer (Figure 5c) demonstrate the effectiveness of our staged curriculum. The optimization proceeds in two clean phases. In Stage 1, the relational loss converges sharply while the object loss decreases more slowly. In Stage 2, the relational loss remains stable, proving the structure is preserved, while the

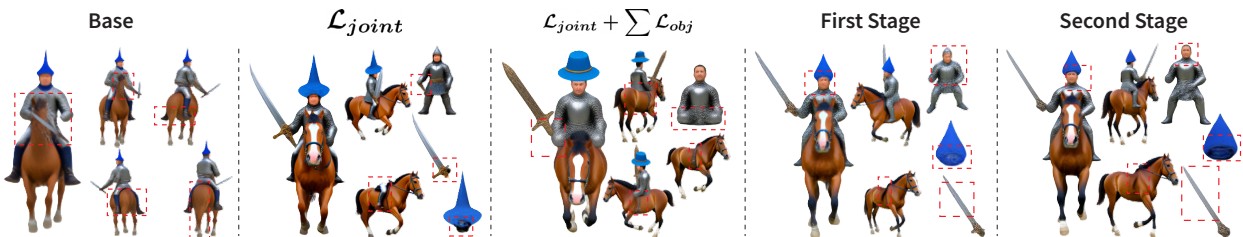

| Base | $\mathcal{L}_{joint}$ | $\mathcal{L}_{joint} + \sum \mathcal{L}_{obj}$ | First Stage | Second Stage |

Figure 7: Results of an ablation study on DecompDreamer with the text prompt: "A knight in full shining armor galloping on a brown horse, wearing a blue hat, and holding a sword". Additional ablation studies can be found in Figure 11 of the appendix.

object loss begins its own rapid descent. This plot is definitive proof of temporal decoupling, which avoids the gradient conflicts that plague other heuristics and ensures a stable convergence.

**Phase-Specific Divergence Rate ($\mathcal{D}_{rate}$).** To rigorously quantify optimization stability and validate the theoretical analysis of gradient conflicts, we introduce the *Phase-Specific Divergence Rate*. We fit a linear regression model to the logarithm of the total loss, $\log(\mathcal{L}_{\text{total}})$, and report the slope. To rigorously quantify optimization stability and validate the theoretical analysis of gradient conflicts, we introduce the *Phase-Specific Divergence Rate*. We fit a linear regression model to the logarithm of the total loss, $\log(\mathcal{L}_{\text{total}})$, and report the slope: $\mathcal{D}_{\text{rate}} = \text{Slope}(\log(\mathcal{L}_{\text{total}}))$. To avoid bias from initialization transients, we exclude the first 10% of iterations. We compute $\mathcal{D}_{\text{rate}}$ separately for the two optimization phases—the *Structure Phase* ($t < \gamma T$) and the *Refinement Phase* ($t \geq \gamma T$). For fair comparison, all baselines are evaluated using the same temporal partitions. A positive slope ($\mathcal{D}_{\text{rate}} > 0$) indicates divergence (rising loss due to gradient conflict), while a negative slope indicates stable convergence.

As summarized in Table 1, all methods exhibit stable convergence during the structure phase ($t < \gamma T$). However, Gala3D and GraphDreamer shows significant divergence ($\mathcal{D}_{\text{rate}} > 0$) during the refinement phase, confirming the "tug-of-war" hypothesis. In contrast, DecompDreamer maintains negative slopes throughout both phases, statistically validating that our staged curriculum resolves the gradient conflict bottleneck and achieves the most stable optimization trajectory.

Table 1: Quantitative analysis of the Divergence Rate metric ($\downarrow$), measuring gradient conflict—lower (more negative) values indicate better stability.

| Method | $t < \gamma T$ | $t \geq \gamma T$ |
|---|---|---|
| GALA3D (H2) | -5.62e-4 | +2.75e-5 |
| GraphDreamer (H3) | -5.64e-5 | +7.60e-5 |
| **Ours (H4)** | **-6.48e-4** | **-7.73e-4** |

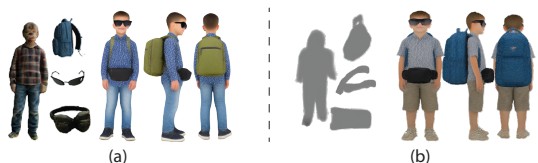

Figure 8: Qualitative comparison of DecompDreamer with different coarse intializers: (a) TripoSR initialization, (b) Point-E initialization.

## 6.2 Comparison to State-of-the-art Methods

We evaluate DecompDreamer against state-of-the-art text-to-3D methods, including Gala3D, GraphDreamer, MVDream, LucidDreamer, Magic3D, Step1X-3D, Hunyuan3D-2.0 and Trellis, across a suite of complex compositional prompts. Although CompGS reports competitive results, it is omitted here due to the lack of publicly available code at the time of experimentation.

**Quantitative Evaluation.** We benchmark all methods using standard quantitative metrics, including CLIP Score Radford et al. (2021), Pick-A-Pic Kirstain et al. (2023), and Text-to-3D Alignment Duggal et al. (2025), over 30 compositional prompts containing 2–11 objects. To more directly assess compositional reasoning and spatial coherence, we further introduce three targeted evaluations:

1. **User Preference Study:** A perceptual study ($N = 20$ participants) comparing overall visual fidelity and prompt alignment.

2. **Geometric Entanglement Rate (GER):** A focused evaluation ($N = 15$ participants) in which participants flag objects that appear geometrically merged, blended, or lacking clear boundaries. GER is defined as the proportion of flagged objects among all generated objects, providing a direct measure of disentanglement.

3. **Relational Fidelity:** An automated Vision-Language Model evaluation (GPT-4o) that verifies adherence to the reference scene graph $G = (V, E)$ from Section 5.1. We report (a) *Object Presence Score* ($\mathcal{S}_{\text{soft}}$), measuring the percentage of prompts where all objects ($V$) are correctly detected, and (b) *Relational Adherence Score* ($\mathcal{S}_{\text{hard}}$), measuring the percentage of prompts where all relational edges ($E$) are satisfied.

The results are summarized in Table 2. DecompDreamer achieves the highest overall performance across all evaluation metrics, with a clear overall user preference of 51%. It attains the top CLIP, Pick-A-Pic, and Text-to-3D Alignment scores, with the latter (73.7%) surpassing the nearest competitor by a substantial margin. Meanwhile, its perfect Relational Fidelity (100%) and low Geometric Entanglement Rate (GER = 3.24%) confirm strong structural coherence and disentanglement. Table 5 in the appendix further divides the results by prompt complexity, distinguishing between scenes with ≤3 and >3 objects. Although some baselines perform competitively on simpler prompts, their performance deteriorates sharply as relational complexity increases; for example, GraphDreamer's alignment score drops from 50.0% to 16.5%. Even in simpler cases, users still prefer DecompDreamer (45.0%), underscoring its consistent advantage. These findings demonstrate that DecompDreamer uniquely scales to complex multi-object scenes without sacrificing spatial accuracy or relational consistency. Additional details are deferred to the Sections A.5, A.6, and A.7 of the appendix.

**Disentanglement in Multi-Object 3D Generation.** Beyond compositional fidelity, achieving geometric disentanglement is critical for high-quality multi-object generation. As shown in Figure 6, both GALA3D and GraphDreamer suffer from object blending and boundary artifacts—particularly when strong inter-object dependencies are present. In contrast, DecompDreamer maintains sharp, well-separated object boundaries through its targeted relational optimization and view-aware refinement stages. These qualitative results are consistent with the low Geometric Entanglement Rate (GER = 3.24%) reported in Table 2, confirming the framework's ability to preserve structural separation while accurately modeling complex relational semantics. Additional qualitative comparisons are provided in Appendix A.8.

Table 2: Comparison of CLIP Score, Pick-A-Pic, Text-to-3D alignment, Relational Fidelity, User Preference Study, and Execution Time. All metrics except execution time are reported as percentages across the full 30-prompt benchmark (2-11 object scenes); execution time is reported in minutes for generating 6 objects.

| Method | CLIP Score ↑ | Pick-A-Pic ↑ | T3D Align. ↑ | Relational Fidelity ↑ | GER ↓ | User Pref. Study ↑ | Exec. Time (min) ↓ |
|---|---|---|---|---|---|---|---|
| Magic3D | 31.8 | 7.0 | 46.3 | 38.3 | 52.01 | 0.0 | 49 |
| GraphDreamer | 28.2 | 8.0 | 38.8 | 53.5 | 41.87 | 9.0 | – |
| CompGS | 30.2 | 9.0 | _62.6_ | _72.3_ | _20.9_ | _10.0_ | – |
| GALA3D | 30.2 | 9.0 | 56.8 | 69.8 | 26.39 | 6.0 | 360 |
| LucidDreamer | _33.0_ | _12.0_ | 46.8 | 32.4 | 72.22 | 3.0 | 45 |
| MVDream | 32.1 | _12.0_ | 38.0 | 47.3 | 57.12 | 0.0 | 43 |
| Step1X-3D | 29.6 | 7.0 | 35.1 | 49.1 | 22.19 | 8.0 | **0.2** |
| Hunyuan3D-2.0 | 29.2 | 8.0 | 34.5 | 48.1 | 22.45 | 6.0 | _0.5_ |
| Trellis | 28.4 | 7.0 | 33.9 | 48.4 | 41.90 | 6.0 | **0.2** |
| **Ours** | **34.7** | **21.0** | **73.7** | **100.0** | **3.24** | **51.0** | 36 |

## 6.3 Ablation Study

We conduct an ablation study to validate our design choices, with results in Figure 7. The Base model, trained without decomposition, fails to produce coherent 3D structures (e.g., a malformed horse). $\mathcal{L}_{\text{joint}}$

optimizes inter-object relationships, resulting in better layout but poor disentanglement. For instance, the 3D object corresponding to the horse includes a leg from the knight. $\mathcal{L}_{\mathrm{joint}} + \sum \mathcal{L}_{\mathrm{obj}}$ applies equal weight to both relational and object-level optimization. However, this confuses the model, often leading to structural inconsistencies where neither relationships nor individual objects are properly generated. For example, in the full composition, the horse contains only a faint leg from the knight, while the individual knight component lacks well-defined limbs. In contrast, our full composition-aware optimization succeeds. The staged curriculum first ensures a coherent relational structure before refining object fidelity, which resolves the conflicts observed in other variants and enhances overall structural integrity. Additional results are in Appendix A.8.

## 6.4 Execution Time Comparison Between Optimization-Based Methods

Table 3 reports the wall-clock time (in minutes) required by each method to generate 3D content as a function of the number of objects in the scene. Our method exhibits a consistent and scalable runtime trend. Across all object counts, it requires approximately 1/10 of the runtime of GALA3D, while simultaneously producing higher-fidelity, geometrically consistent, and semantically accurate compositions. In contrast, GraphDreamer maintains a constant generation time of 180 minutes but fails to scale beyond four objects, frequently breaking down as relational complexity increases. This behavior highlights the instability of its iterative optimization schedule, which struggles to converge in high-dimensional compositional settings despite substantial computation time.

Methods such as LucidDreamer, Magic3D, MVDream, and Trellis exhibit nearly constant and fast generation times (ranging from 0.2 to 49 minutes); however, these approaches are primarily designed for single-object synthesis. While they can occasionally handle two-object scenes and sometimes succeed with three objects, they consistently fail to capture complex inter-object relationships and compositional semantics once the number of entities exceeds this threshold. Overall, our method is the only approach that scales gracefully in both visual fidelity and semantic alignment as scene complexity increases, while remaining significantly faster than GALA3D and substantially more robust than GraphDreamer at higher object counts.

Table 3: Time (in minutes) comparison of different methods across varying number of objects.

| Method | #2 | #3 | #4 | #11 |
|---|---|---|---|---|
| Magic3D | 49 | 49 | 49 | 49 |
| GraphDreamer | 180 | 180 | 180 | - |
| GALA3D | 120 | 180 | 240 | 660 |
| LucidDreamer | 45 | 45 | 45 | 45 |
| MVDream | 43 | 43 | 43 | 43 |
| Step1X-3D | 0.2 | 0.2 | 0.2 | 0.2 |
| Hunyuan3D-2.0 | 0.5 | 0.5 | 0.5 | 0.5 |
| Trellis | 0.2 | 0.2 | 0.2 | 0.2 |
| **Ours** | 12 | 18 | 24 | 66 |

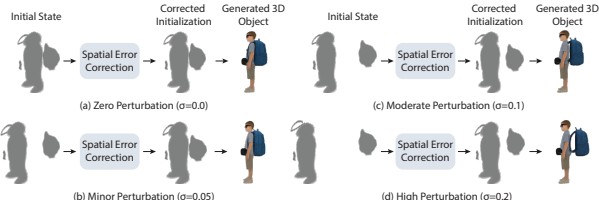

Figure 9: Ablation study of the robustness of the Spatial Error Correction Module to Gaussian perturbations applied to predicted object centers ($\sigma \in \{0.0, 0.05, 0.1, 0.2\}$). Even under severe misalignment ($\sigma = 0.2$), the module successfully realigns objects prior to final generation.

## 6.5 Robustness to Noisy Spatial Priors

To evaluate the system's resilience to incorrect spatial priors or noisy VLM outputs, we conducted a stress test by manually injecting Gaussian noise into the predicted object centers. As illustrated in Figure 9, we applied perturbations of increasing magnitude ($\sigma \in \{0.0, 0.05, 0.1, 0.2\}$) to the initial layouts.

The Spatial Error Correction Module (Section 5.2.3) is critical for handling these errors. By freezing object geometry and optimizing only the translation parameters ($T^o$) using the joint edge loss ($\mathcal{L}_{joint}$), the system allows diffusion priors to physically "pull" objects into their correct semantic interaction zones. As shown in the "Corrected Initialization" column of Figure 9, the module successfully corrects significant misalignments, even at high noise levels ($\sigma = 0.2$), effectively "snapping" the backpack onto the figure's back before fine-grained refinement begins.

## 6.6 Effect of Different Coarse Initializers

While our default framework utilizes TripoSR Tochilkin et al. (2024), we validate that our method's high fidelity is not dependent on this specific initialization. We replace TripoSR with Point-E Nichol et al. (2022) and observe that while the generation time increases significantly (7.5× slowdown, or roughly 180 minutes vs. 24 minutes), the visual quality remains consistent (Figure 8). This confirms that the superior fidelity of DecompDreamer stems from the proposed optimization curriculum rather than the initialization prior. Additionally, to ensure a fair comparison with prior compositional methods that rely on Point-E, we benchmark using identical initialization (Figure 21). Even under these constrained conditions, DecompDreamer significantly outperforms baselines in preserving object geometry and relational structure while requiring comparable runtime.

## 7 Conclusion and Future Works

In this paper, we reframed compositional text-to-3D generation as a problem of optimization scheduling, arguing that the failures of prior methods are predictable outcomes of heuristics that collapse under a combinatorial explosion of conflicting gradients. We introduced DecompDreamer, which addresses this theoretical bottleneck with a staged, composition-aware optimization strategy. Functioning as an implicit curriculum, our method temporally decouples competing objectives by first establishing a coherent relational scaffold before refining the high-fidelity details of individual objects. Our extensive experiments validate this approach, demonstrating state-of-the-art performance in fidelity, disentanglement, and spatial coherence, especially on complex scenes where other heuristics fail. While this work focuses on generating assets composed of distinct object types, a promising future direction is the decomposition of a single entity into finer-grained subcomponents—such as segmenting a 3D human model into regions like the face, hands, or limbs. In such settings, DecompDreamer's composition-aware optimization could serve as a foundational strategy. Addressing these challenges may require improved representations for fine-grained parts, which we leave as an exciting avenue for future work.

## 8 Acknowledgement

This work was supported by NSF grant 2323086.

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

# A   Appendix

## A.1   Additional Loss Dynamics Analysis

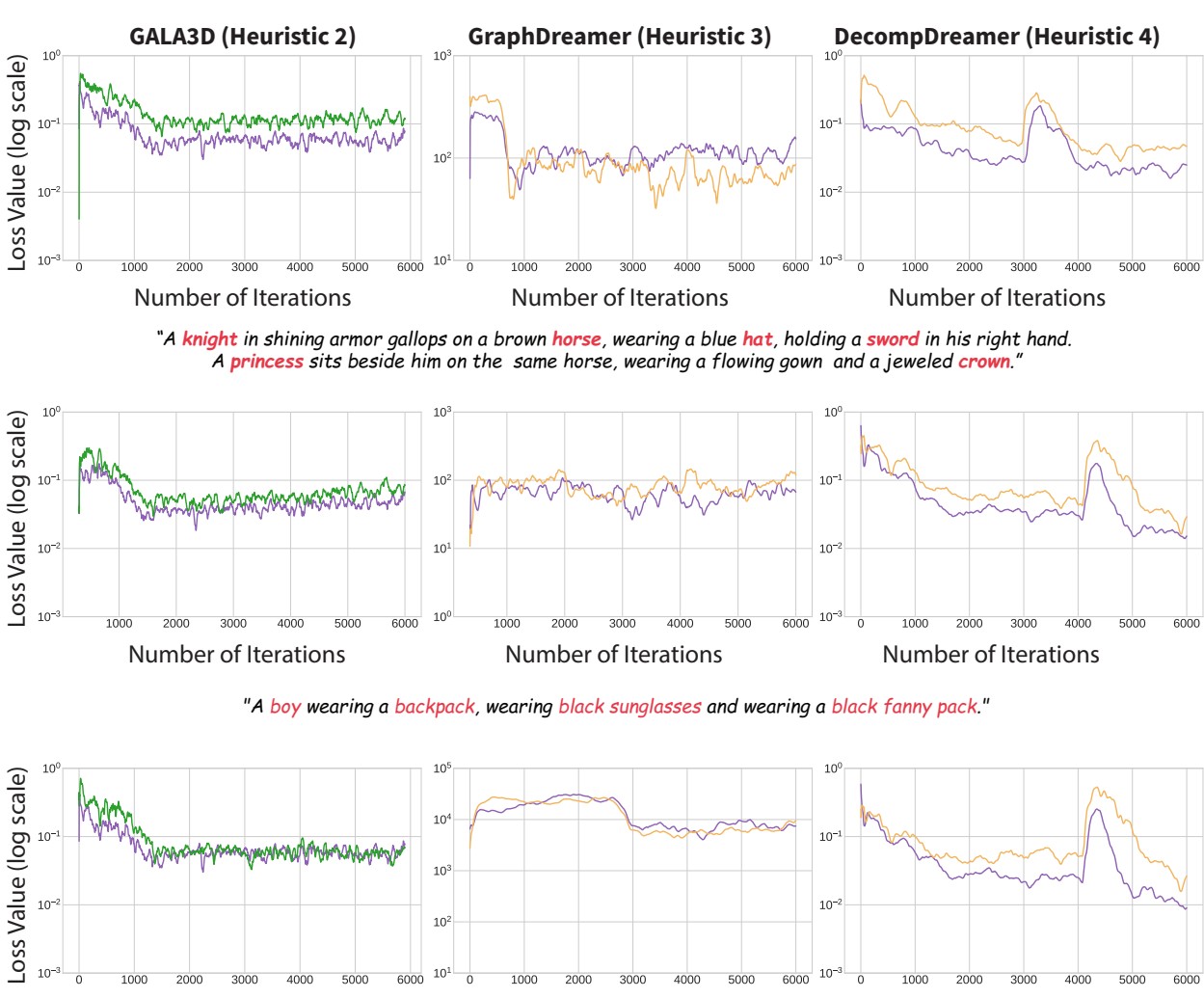

Figure 10: Additional empirical comparison of optimization heuristics across three complex text prompts (rows). We plot individual loss components (log scale) over iterations. The columns illustrate that GALA3D (Heuristic 2) converges suboptimally with high remaining loss, GraphDreamer (Heuristic 3) exhibits significant instability and divergence, while our staged approach, DecompDreamer (Heuristic 4), achieves stable convergence to lower loss minima.

### A.2 Notations and Algorithm

Table 4: Summary of Notations Used in the Algorithm

| Notation | Description |
|:---:|:---|
| $t^g$ | Compositional text prompt |
| VLM | Vision-Language Model |
| $T_{\max}$ | Total number of optimization iterations |
| $\mathcal{I}$ | Generated image representation |
| $G$ | Scene graph extracted from VLM |
| $N$ | Number of objects in the scene |
| $E$ | Edge list (connections between objects) |
| $O$ | Set of all objects in the scene |
| $t_i^o$ | Text prompt for object $O_i$ |
| $t_{ij}^e$ | Text prompt for edge $(i,j)$ |
| $(X_i, Y_i, Z_i)$ | 3D position of object $O_i$ |
| $s_i$ | Scale factor for object $O_i$ |
| $T^O$ | Translation Parameters |
| $\theta_i$ | Rotation parameters for object $O_i$ |
| $\mathbf{R}(\theta_i)$ | Rotation matrix for object $O_i$ |
| $\mathbf{T}(X_i, Y_i, Z_i)$ | Translation matrix for object $O_i$ |
| $\text{Init3D}(t_i^o)$ | Coarse 3D Initializer for object $O_i$ |
| $P(O_i)$ | Point cloud representation of object $O_i$ |
| $P(O_i)_{\text{aligned}}$ | Aligned point cloud of $O_i$ after transformations |
| $\mathcal{G}_{o_i}$ | Gaussian representation of object $O_i$ |
| $\mathcal{G}$ | Combined Gaussian representation of the scene |
| $n_e$ | Number of edges in the scene graph ($|E|$) |
| $\mathcal{L}_{\text{joint}}$ | Loss for Joint Optimization |
| $\mathcal{L}_{\text{obj}}$ | Object-specific SDS loss |
| $\mathcal{L}_{\text{scene}}$ | Scene-level loss |
| $\mathcal{L}_{\text{target}}$ | Loss for targeted Optimization |
| $\mathcal{L}_{\text{SDS}}$ | Score Distillation Sampling loss |
| $\mathcal{L}_{\text{loss}}$ | Total loss used for optimization |
| $\lambda$ | Weight factor to control contribution of $\mathcal{L}_{obj}$ |
| $\gamma$ | Weight controlling the phase transition in overall optimization |
| $z_{ij}$ | Rasterized and encoded representation of gaussians |
| Adam() | Adam optimizer for updating Gaussian parameters |
| DensifyAndPrune() | Function to densify and prune Gaussian representations |
| IsRefinementIteration($t$) | Function to check if densification is needed at iteration $t$ |

---

**Algorithm 1** Structured Decomposition and Composition-Aware Optimization

---

**Input:** $t^g$: Compositional text prompt, VLM: Vision-Language Model, $T_{\max}$: Total iterations

1: **Stage 1: Structured Decomposition and Initialization**
2: $\mathcal{I}, G, N \leftarrow \text{VLM}(t^g)$         ▷ Generate image and scene graph
3: $E, O, \{t_i^o, t_{ij}^e, X_i, Y_i, Z_i, s_i, \theta_i\} \leftarrow \text{VLM}(\mathcal{I}, G)$
4: **for** $i = 1$ to $N$ **do**         ▷ Convert objects to Gaussians
5:     $P(O_i) \leftarrow \text{Init3D}(t_i^o)$
6:     $P(O_i)_{\text{aligned}} \leftarrow s_i \cdot \mathbf{R}(\theta_i) \cdot P(O_i) + \mathbf{T}(X_i, Y_i, Z_i)$
7:     $\mathcal{G}_{o_i} \leftarrow P(O_i)_{\text{aligned}}$
8: **end for**
9: $\mathcal{G} = \bigcup_{i=1}^{N} \mathcal{G}_{o_i}$         ▷ Unified Gaussian representation
10: $T^O = \bigcup_{i=1}^{N}(X_i, Y_i, Z_i)$         ▷ 3D positions as Translation parameters
11: **for** $t = 1$ to $T_{\text{translation}}$ **do**
12:     $\mathcal{L}_{\text{spatial-error-correction}} \leftarrow \sum_{(i,j) \in E} \nabla_{T_i^O, T_j^O} \mathcal{L}_{\text{SDS}}(z; y_{ij}^e)$
13:     $T^O \leftarrow \text{Adam}(\nabla \mathcal{L}_{\text{spatial-error-correction}})$         ▷ Update step
14: **end for**
15: **Stage 2: Composition-Aware Optimization**
16: $n_e \leftarrow |E|$         ▷ Number of edges
17: **for** $t = 1$ to $T_{\max}$ **do**
18:     **if** $t < \gamma.T_{\max}$ **then**         ▷ Joint Optimization
19:         **if** $t \bmod (n_e + 1) \neq 0$ **then**
20:             $i, j \leftarrow E[t \bmod n_e]$
21:             $\mathcal{G}'_{o_i} \leftarrow \mathcal{G}_{o_i} + T_i^o, \quad \mathcal{G}'_{o_j} \leftarrow \mathcal{G}_{o_j} + T_j^o$
22:             $\mathcal{L}_{\text{joint}} \leftarrow \nabla_{\mathcal{G}'_{o_i}, \mathcal{G}'_{o_j}} \mathcal{L}_{\text{SDS}}(z; y_{ij}^e)$
23:             $\mathcal{L}_{\text{obj}} \leftarrow \sum_{O \in \{O_i, O_j\}} \nabla_{\mathcal{G}_O} \mathcal{L}_{\text{SDS}}(z; y_{a_O}^o)$
24:             $\mathcal{L}_{\text{loss}} \leftarrow \mathcal{L}_{\text{joint}} + \lambda \left(\frac{t}{T_{\max}}\right)^2 \mathcal{L}_{\text{obj}}$
25:         **else**
26:             **for** $i = 1$ to $N$ **do**
27:                 $\mathcal{G}'_{o_i} \leftarrow \mathcal{G}_{o_i} + T_i^o$
28:             **end for**
29:             $\mathcal{L}_{\text{scene}} \leftarrow \nabla_{\mathcal{G}} \mathcal{L}_{\text{SDS}}(z; y^g)$
30:             $\mathcal{L}_{\text{loss}} \leftarrow \mathcal{L}_{\text{scene}}$
31:         **end if**
32:     **else**         ▷ Targeted Optimization
33:         $i, j \leftarrow E[t \bmod n_e]$
34:         $\mathcal{L}_{\text{target}} \leftarrow \sum_{O \in \{O_i, O_j\}} \nabla_{\mathcal{G}_O} \mathcal{L}_{\text{SDS}}(z; y_{ij}^e)$
35:         $\mathcal{L}_{\text{obj}} \leftarrow \sum_{O \in \{O_i, O_j\}} \nabla_{\mathcal{G}_O} \mathcal{L}_{\text{SDS}}(z; y_{a_O}^o)$
36:         $\mathcal{L}_{\text{loss}} \leftarrow \mathcal{L}_{\text{target}} + \mathcal{L}_{\text{obj}}$
37:     **end if**
38:     $\mathcal{G} \leftarrow \text{Adam}(\nabla \mathcal{L}_{\text{loss}})$         ▷ Update step
39:     **if** IsRefinementIteration($t$) **then**
40:         $\mathcal{G} \leftarrow \text{DensifyAndPrune}(\mathcal{G})$
41:     **end if**
42: **end for**

---

## A.3 Hyperparameters

Our optimization framework follows the learning rates and camera parameter updates used in LucidDreamer Liang et al. (2024). Each object is allocated 250 iterations, with the first 12 iterations serving as a warm-up.

Densification and pruning begin after 50 iterations and continue until 125 iterations, with a 25-iteration interval for both operations. The azimuth angle is constrained to [-180, 180] to ensure stable optimization. The minimum timestep is 0.02, while the maximum timestep, initially 0.98, is annealed to 0.5 during warm-up and remains constant thereafter. The hyperparameters $\lambda$ and $\gamma$ in Eq. 6 are set to 8 and 0.6, respectively. The warm-up phase is restarted in the second stage for 30% of the remaining iterations.

### A.4 Coarse Initialization Details

Each object $O_i$ is initialized with an object-specific point cloud $P(O_i)$ generated using TripoSR Tochilkin et al. (2024). This initialization provides a coarse geometric prior for each object before compositional optimization.

To ensure correct placement and orientation within the scene, each point cloud is aligned using spatial attributes inferred by the Vision–Language Model (VLM). Specifically, the VLM predicts the object center coordinates $(x_i, y_i, z_i)$, a relative scale factor $s_i$, and an orientation parameter $\theta_i$ for object $O_i$. These attributes are used to transform the corresponding point cloud through a sequence of scaling, rotation, and translation operations, yielding an aligned object representation:

$$P(O_i)_{\text{aligned}} = s_i \cdot R(\theta_i) \cdot P(O_i) + T(x_i, y_i, z_i),$$

where $R(\theta_i)$ denotes the rotation matrix derived from the predicted orientation and $T(x_i, y_i, z_i)$ denotes the translation operator defined by the predicted object center.

The full scene is then constructed by aggregating the aligned point clouds of all objects:

$$P_{\text{scene}} = \bigcup_{i=1}^{n} P(O_i)_{\text{aligned}}.$$

For subsequent optimization, each aligned point cloud is converted into a set of 3D Gaussians. To enable independent and disentangled refinement, we explicitly maintain a one-to-one association between Gaussians and objects. Formally, the scene representation is expressed as:

$$\mathcal{G} = \bigcup_{i=1}^{n} \mathcal{G}_{o_i},$$

where $\mathcal{G}_{o_i}$ denotes the subset of Gaussians derived from $P(O_i)_{\text{aligned}}$. This object-level partitioning allows targeted optimization of geometric and appearance attributes for individual objects, while preserving clear boundaries and preventing unintended feature blending during compositional refinement.

### A.5 User Study Protocol

We conducted a randomized, double-blind user preference study over 30 prompts ($N = 20$ evaluators). For each prompt, evaluators viewed videos from our method and all baselines in a randomized order and selected the result that best matched the prompt based on visual quality and semantic alignment. Method identities were anonymized to prevent bias. We instructed participants: "Please watch each set of videos and select the one you think best matches the given prompt. Please base your selection on both image quality and semantic alignment."

### A.6 Quantifying Optimization Stability and Failure Modes

**Geometric Entanglement Rate (GER).** To differentiate between clean disentanglement and the "feature blending" caused by simultaneous optimization, we conducted a focused user study ($N = 15$). Unlike standard preference studies, evaluators were tasked with a binary classification failure analysis. For each generated scene, users identified specific objects that were "physically merged, blended, or lacked distinct geometric boundaries" with respect to their neighbors. The GER is calculated as:

$$\text{GER} = \frac{\text{Total Objects Flagged as Entangled}}{\text{Total Objects Generated}} \times 100\% \tag{7}$$

This metric provides a direct measure of the geometric isolation achieved by the optimization schedule.

**Relational Fidelity Scores.** To assess the model's ability to adhere to complex constraints without "relational violation," we employ an automated VLM-based evaluation using GPT-4o. To ensure the evaluation strictly reflects the compositional structure of the input prompt, we derive our validation criteria directly from the reference scene graph $G = (V, E)$ generated by the VLM (as described in Section 5.1). We map the graph components to two tiers of boolean constraints:

- **Object Presence Score ($\mathcal{S}_{soft}$):** Derived from the set of nodes $V$. For every object $O_i \in V$, we verify its existence in the rendered scene (e.g., "Is there an astronaut?"). The score is the percentage of prompts where *all* nodes are successfully detected.

- **Relational Adherence Score ($\mathcal{S}_{hard}$):** Derived from the set of edges $E$. For every relationship $(O_i, O_j, r) \in E$, we verify the specific interaction (e.g., "Is the astronaut *riding* the horse?"). The score is the percentage of prompts where *all* edges are satisfied, verifying that the model has resolved the combinatorial complexity of the scene.

Table 5: Comparison of CLIP Score, Pick-A-Pic, Text-to-3D Alignment, Relational Fidelity, Disentanglement, and User Preference Study. All values are percentages.

| Method | CLIP Score ↑ | | Pick-A-Pic ↑ | | T3D Align. ↑ | | User Study ↑ | | GER ↓ | | Rel. Fid. ↑ | |
|---|---|---|---|---|---|---|---|---|---|---|---|---|
| # of Objects | ≤3 | >3 | ≤3 | >3 | ≤3 | >3 | ≤3 | >3 | ≤3 | >3 | ≤3 | >3 |
| Magic3D | 32.9 | 29.5 | 7.0 | 8.0 | 59.6 | 19.6 | 0.0 | 0.0 | 26.67 | 83.69 | 45.16 | 49.03 |
| GraphDreamer | 32.6 | 19.4 | 9.0 | 7.0 | 50.0 | 16.5 | 10.0 | 7.0 | 28.33 | 58.78 | 57.43 | 59.96 |
| CompGS | 30.6 | 29.3 | 9.0 | 8.0 | 65.6 | 56.7 | 12.0 | 7.0 | 8.10 | 36.90 | 78.12 | 76.87 |
| GALA3D | 30.3 | 30.1 | 10.0 | 7.0 | 59.3 | 51.8 | 6.0 | 6.0 | 15.00 | 40.63 | 77.89 | 74.79 |
| LucidDreamer | 32.8 | 33.5 | 11.0 | 13.0 | 59.6 | 21.3 | 4.0 | 0.0 | 55.00 | 93.75 | 50.10 | 50.09 |
| MVDream | 32.0 | 32.3 | 12.0 | 11.0 | 48.9 | 16.1 | 0.0 | 0.0 | 46.67 | 70.20 | 40.46 | 43.74 |
| Step1X-3D | 29.5 | 29.7 | 8.0 | 6.0 | 42.5 | 20.2 | 10.0 | 4.0 | 10.00 | 37.42 | 42.33 | 51.35 |
| Hunyuan3D-2.0 | 29.7 | 28.3 | 9.0 | 7.0 | 41.8 | 19.9 | 6.0 | 7.0 | 13.33 | 33.85 | 45.76 | 52.45 |
| Trellis | 28.7 | 27.7 | 7.0 | 6.0 | 41.0 | 19.7 | 7.0 | 5.0 | 43.33 | 40.10 | 38.85 | 45.65 |
| **Ours** | **34.7** | **34.6** | **18.0** | **27.0** | **67.2** | **86.6** | **45.0** | **64.0** | **3.33** | **3.13** | **100.0** | **100.0** |

## A.7 Quantitative Comparison and Discussion

To evaluate the performance of DecompDreamer, we employ four complementary metrics: CLIP Score Radford et al. (2021), Pick-a-Pic Kirstain et al. (2023), Text-to-3D Alignment Duggal et al. (2025), and a user study, applied across 30 prompts containing 2–11 objects. CLIP Score measures the semantic similarity between rendered images and input prompts using vision-language embeddings, providing a proxy for text-image alignment. Pick-a-Pic is a learned perceptual metric that models human preferences between image pairs conditioned on a prompt, capturing fidelity and relevance. Text-to-3D Alignment quantitatively assesses object presence, count, and spatial relations using grounding-based techniques over rendered views. The user study collects human judgments on object identity, relational correctness, and overall scene plausibility. As shown in Table 2, DecompDreamer achieves the highest scores across all four metrics. It leads in CLIP Score (34.5%) and Pick-a-Pic accuracy (24.0%), with strong generalization to complex scenes. It also attains the best Text-to-3D Alignment score (73.4%) and is preferred by users in 61.0% of cases overall, reaching 73.0% in high-object-count settings. These results highlight DecompDreamer's ability to generate semantically faithful, perceptually preferred, and structurally coherent 3D content.

While DecompDreamer achieves the highest scores across all automated metrics, the observed performance gap—typically around 5%—is narrower than expected. This is largely attributable to the limitations of current evaluation metrics in capturing fine-grained geometric and relational fidelity in 3D assets. CLIP Score and Pick-a-Pic treat each rendered image independently and compare it to the same text prompt, without accounting for the viewpoint from which the image was rendered. As a result, assets with geometric inconsistencies—such as those suffering from the Janus problem—can receive artificially high scores, especially from favorable viewpoints. Although Text-to-3D Alignment is designed for 3D evaluation, it still struggles in

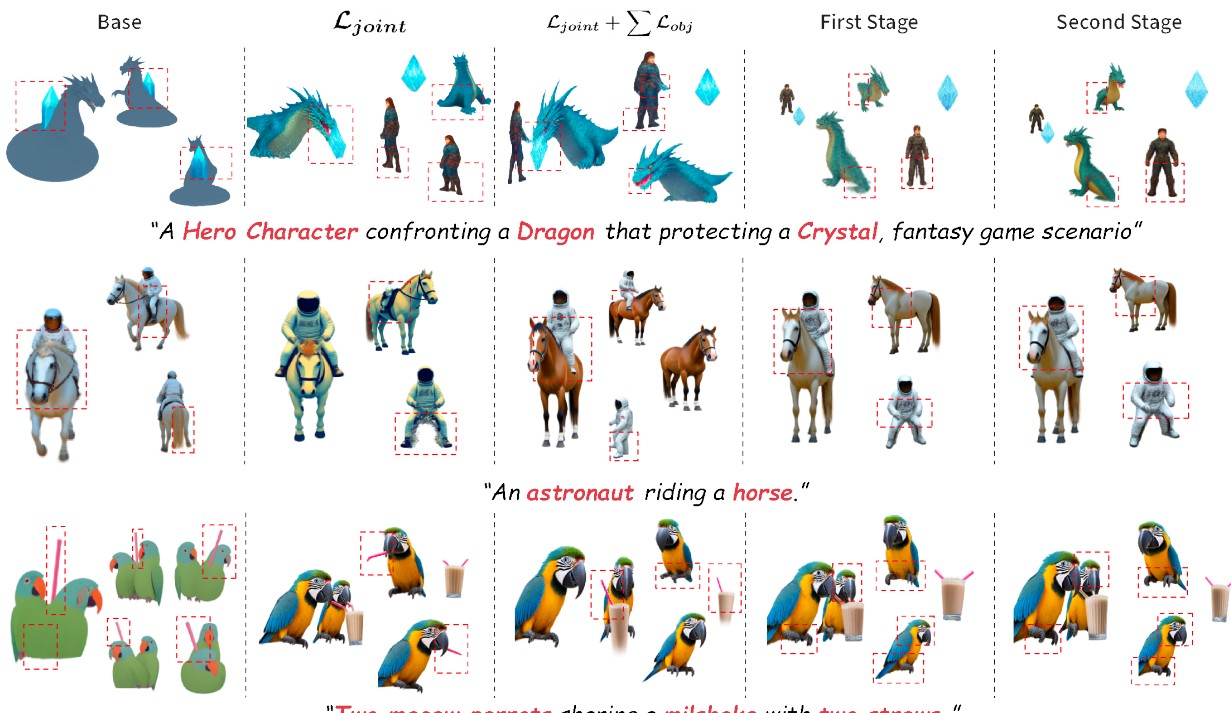

Figure 11: Results of an ablation study on DecompDreamer under different training configurations.

edge cases where outputs are close to the prompt but miss key compositional or relational details. In contrast, the user study, which involves human assessment, is better equipped to capture these nuances, providing a more reliable measure of visual, structural, and semantic fidelity across complex scenes.

### A.8   Additional Ablation Studies

**Training Routine** In Section 6.3, we analyze the impact of different training configurations on compositional 3D generation. Here, we provide additional results and insights. As observed in Figure 11, optimizing only relationships ($\mathcal{L}_{joint}$) produces a well-formed structure but leads to object entanglement, as boundaries between objects remain unclear. On the other hand, when relationships and objects are weighted equally ($\mathcal{L}_{joint} + \sum \mathcal{L}_{obj}$), the model struggles to properly distinguish individual elements, leading to inconsistent object representations. For instance, an astronaut riding a horse appears structurally different from a standalone astronaut, confusing the model and resulting in poor relationship modeling. Our training routine, which progressively transitions from relationship optimization to targeted object refinement, mitigates these issues. The first stage ensures that inter-object interactions are well-established, while the second stage improves individual object details while preserving these relationships.

To disentangle the impact of the proposed staged "structure-then-detail" optimization schedule from other auxiliary components, we conduct an extended ablation study summarized in Table 6. All models are trained under identical conditions using 30 compositional prompts (2–11 objects). Each configuration isolates a distinct factor while holding all others constant, and is conceptually aligned with the heuristic taxonomy introduced in Section 2.2.

**Experimental Variants.**

- Baseline (SD 2.1 only): Standard diffusion guidance without auxiliary components. This configuration is conceptually analogous to *Heuristic 1: Holistic Optimization.*

- Enhanced Baseline (SD 3.5 + Auxiliary Techniques): Incorporates stronger guidance (SD 3.5), view-aware alignment, and negative prompts, while optimizing all objectives simultaneously. This corresponds to *Heuristic 2: Simultaneous Decomposed Optimization.*

- Enhanced + Iterative Optimization: Same setup as above, but with alternating iterative updates rather than joint optimization. This configuration corresponds to *Heuristic 3: Iterative Decomposed Optimization.*

- Baseline + Staged Curriculum: Adds only the proposed two-stage "structure-then-detail" schedule to the Baseline (SD 2.1), without any auxiliary components. This isolates the contribution of the scheduling mechanism itself and is directly aligned with the principles of *Heuristic 4: Staged Decomposed Optimization.*

- DecompDreamer (Full Model): Combines the staged curriculum with SD 3.5 flow-based guidance and all disentanglement-oriented refinement modules.

Table 6 reports CLIP Score, Pick-A-Pic, Text-to-3D Alignment, and Relational Fidelity across different object-count ranges. The *Baseline + Staged Curriculum* configuration alone without newer guidance or refinement techniques, achieves significant gains over both simultaneous and iterative baselines, demonstrating that the scheduling mechanism is a primary driver of convergence stability and relational consistency. The full DecompDreamer model yields the highest overall performance, showing that the curriculum provides a foundation upon which auxiliary techniques further improve local fidelity and disentanglement.

Table 6: Comparison of CLIP Score, Pick-A-Pic, Text-to-3D Alignment, and Relational Fidelity across different object count ranges. All values are percentages.

| Method | CLIP Score ↑ | | | Pick-A-Pic ↑ | | | Text-to-3D Alignment ↑ | | | Relational Fidelity ↑ | | |
|---|---|---|---|---|---|---|---|---|---|---|---|---|
| # of Objects | ≤3 | >3 | All | ≤3 | >3 | All | ≤3 | >3 | All | Soft (Nodes) | Hard (Edges) | Overall |
| Baseline (SD 2.1 only) | 28.9 | 28.1 | 28.6 | 11.0 | 6.0 | 9.0 | 56.4 | 25.2 | 46.0 | 60.3 | 36.5 | 48.4 |
| Enhanced (SD 3.5 + Aux Techniques) | 30.5 | 30.2 | 30.4 | 13.0 | 8.0 | 11.0 | 59.3 | 51.0 | 56.5 | 77.4 | 59.6 | 68.5 |
| Enhanced + Iterative Optimization | 32.8 | 20.3 | 28.6 | 14.0 | 10.0 | 12.0 | 61.9 | 56.7 | 60.2 | 72.1 | 65.7 | 68.9 |
| Baseline + Staged Curriculum | 33.1 | 32.9 | 33.0 | 30.0 | 36.0 | 33.0 | 66.7 | 82.2 | 71.9 | 100.0 | 100.0 | 100.0 |
| DecompDreamer (Full Model) | 34.7 | 34.6 | 34.7 | 32.0 | 40.0 | 36.0 | 67.2 | 86.6 | 73.7 | 100.0 | 100.0 | 100.0 |

**Azimuth Adjustment** We ablate the impact of azimuth correction and object-view-dependent embeddings in Eq. equation 4. Without azimuth correction, objects exhibit noticeable disorientation. As illustrated in Figure 12, omitting object-view adjustments in the *robot* example leads to misplaced features—for instance, the robot's eyes appear on the back of its head. This misalignment arises because the global front view of the entire composition does not align with the object's canonical front. As a result, the losses $\mathcal{L}_{\text{obj}}$ and $\mathcal{L}_{\text{joint}}$ may optimize in conflicting directions, hindering convergence and structural coherence.

**Negative Prompt** In Section 5.2.2, we discuss the use of negative prompts to enhance object disentanglement. Figure 13 presents experiments comparing results with and without negative prompts. For instance, in the case of the Panda, which is connected to a chair, hat, and ficus, DecompDreamer includes the Panda in the negative prompt for each connected object. As a result, there is negligible instance leakage of the Panda onto other objects. In contrast, without negative prompts, Panda fur artifacts appear on the chair. Similar effects are observed in other prompts, demonstrating the effectiveness of negative prompts in maintaining clear object boundaries.

**Spatial Error Correction** We conduct an ablation study to evaluate the impact of spatial error correction, as discussed in Section 5.2.3. Figure 14 compares results generated with and without error correction. Without error correction, objects often fail to fully conform to their intended inter-object relationships, leading to spatial inconsistencies.

**Comparison with Stable Diffusion 2.1** We further analyze the impact of using SDS-based SD2.1 versus SDS-based SD3.5 in DecompDreamer. While SDS-based SD2.1 outperforms other SOTA methods

**w/o Azimuth adjustment**          **w/ Azimuth adjustment**

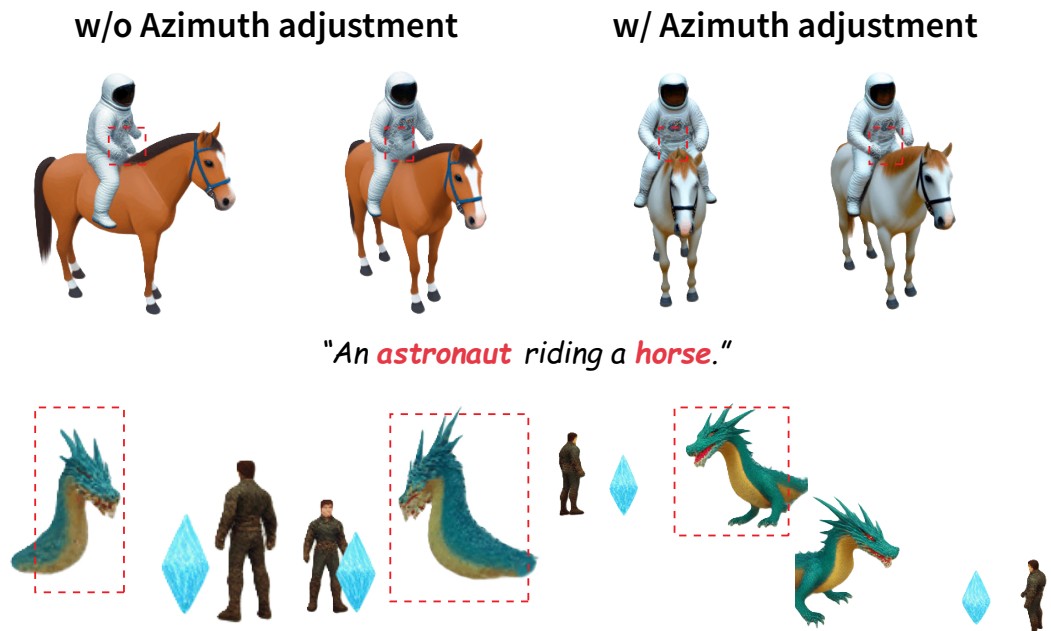

*"An **astronaut** riding a **horse**."*

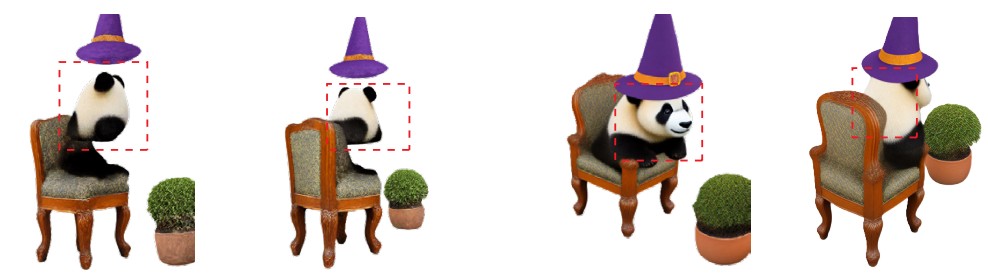

*"A **Hero Character** confronting a **Dragon** that protecting a **Crystal**, fantasy game scenario"*

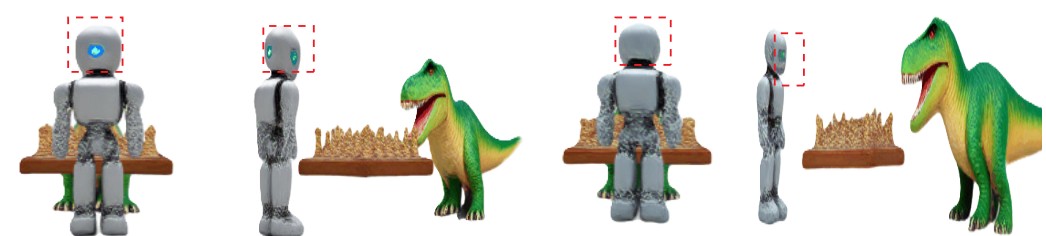

*"**Panda** in a **wizard hat** sitting on a wooden **chair** and looking at a **ficus in a pot**."*

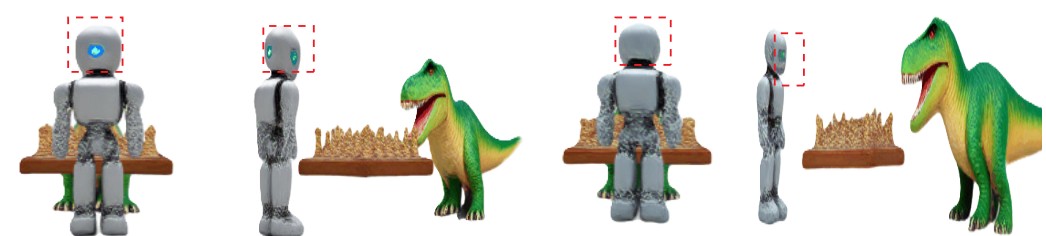

*"a DSLR photo of a **robot** and **dinosaur** playing **chess**."*

Figure 12: Results of an ablation study on DecompDreamer with (right) and without (left) the azimuth adjustment for object-view dependent text prompts.

in relationship modeling, it still struggles with sharpness and convergence speed. By replacing SD2.1 with SD3.5, we significantly improve object detail and reduce optimization time, requiring only one-fourth of the iterations. As shown in Figure 22, SD3.5 produces sharper edges and more refined object details compared to SD2.1.

**Disentanglement in Multi-Object Generation** Here, we present additional results highlighting that, beyond generating compositional 3D scenes, producing high-quality, disentangled individual objects is equally critical. Figure 15 compares individual objects generated by DecompDreamer, GALA3D, and GraphDreamer

Figure 13: Results of an ablation study on DecompDreamer with (right) and without (left) the negative prompts for the objects

on complex prompts. Both GALA3D and GraphDreamer exhibit poor disentanglement, with GraphDreamer being particularly prone to object blending due to its stronger emphasis on relationship modeling. In contrast, DecompDreamer achieves superior disentanglement through targeted relationship optimization and view-aware object refinement.

## A.9 Comparison with Layout-your-3D

Figure 16 compares our method against Layout-your-3D on the prompt *"A lego man riding a horse, the horse is wearing a blue hat."* Layout-your-3D relies on simultaneous optimization constrained by rigid 2D blueprint priors. As hypothesized in our discussion on Heuristic 2, this approach limits the model's ability to adapt geometry for complex interactions. Consequently, the model fails to disentangle the components: the **blue hat merges with the horse's head**, and the **Lego man is entangled with the horse's body**. Conversely, DecompDreamer employs a staged curriculum that successfully separates these entities, resolving the physical dependencies to produce a coherent composition.

## A.10 Comparison with Feed-Forward Methods

Figure 17 compares DecompDreamer against state-of-the-art feed-forward models. While methods like Step1X-3D and Hunyuan3D-2.0 produce high-fidelity geometry, they suffer from significant object fusion in complex scenes, treating multi-object prompts (e.g., a knight riding a horse) as a single monolithic mesh. Similarly, PartCrafter struggles with global coherence, generating disconnected or semantically unrelated components. In contrast, DecompDreamer utilizes staged optimization to enforce explicit disentanglement, preserving the individuality and relational validity of each object.

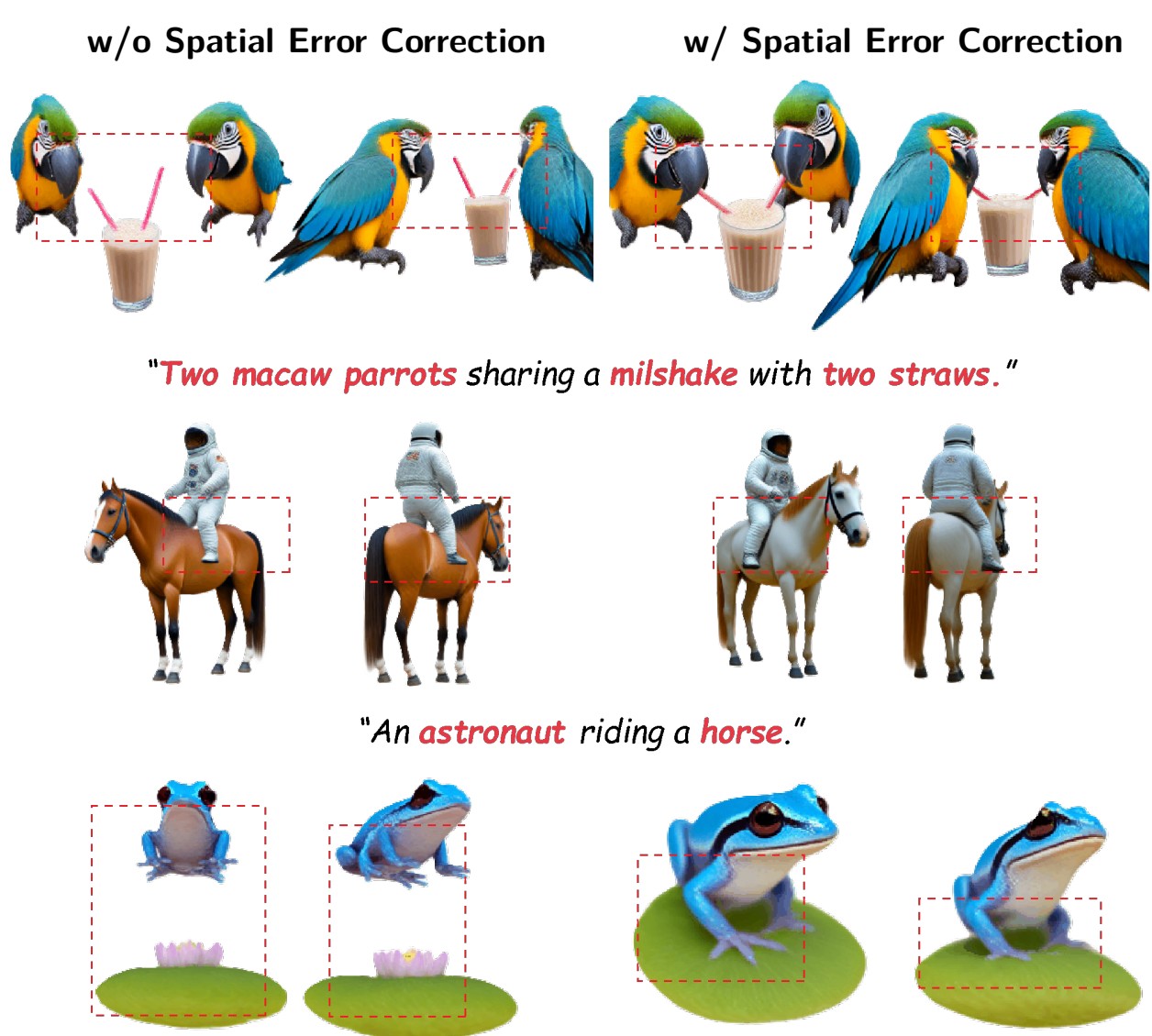

Figure 14: Results of an ablation study on DecompDreamer with (right) and without (left) spatial error correction.

### A.11 Sample usage of ChatGPT

In Figure 18, we showcase how ChatGPT can be leveraged to generate compositional 3D objects using DecompDreamer. Given a user-defined prompt, ChatGPT generates an image and a scene graph, determining the total number of objects. It then constructs text prompts for objects, edges, and corresponding negative prompts. Next, spatial geometry is estimated, followed by 3D object initialization. Finally, the canonical 0-degree view is provided to ChatGPT to estimate the azimuth offset, ensuring accurate object orientation.

### A.12 More Comparisons

In this section, we present additional qualitative comparisons of DecompDreamer against GALA3D, Graph-Dreamer, MVDream, LucidDreamer, Magic3D, Trellis and HiFA. In Figures 23, 24 and 25, our results show

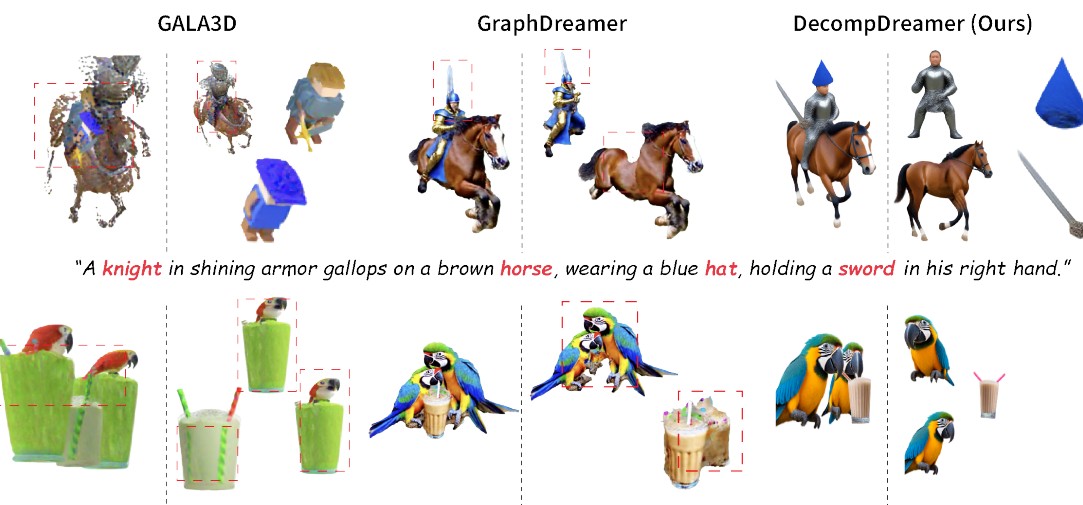

Figure 15: More qualitative comparison of object disentanglement in GALA3D, GraphDreamer, and Decomp-Dreamer. Each method displays the full 3D object (left) and its individual components (right).

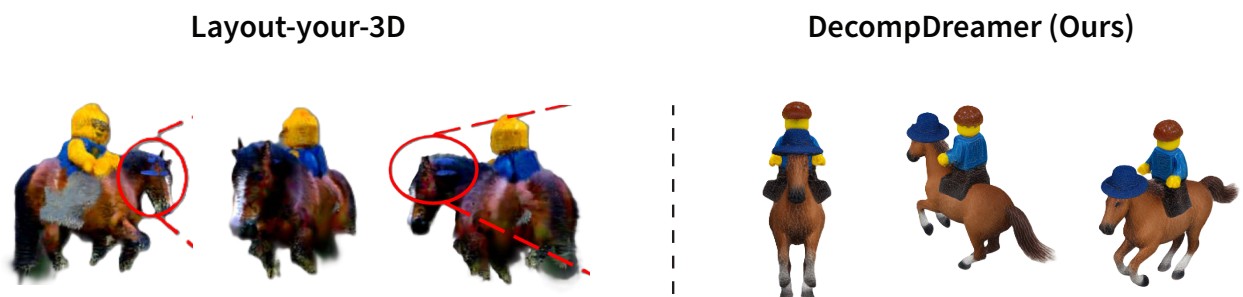

Figure 16: Qualitative comparison on the prompt: 'A lego man riding a horse, the horse is wearing a blue hat.' Layout-your-3D merges the hat with the horse's head and entangles the rider within the body. In contrast, DecompDreamer successfully resolves these physical interactions.

that DecompDreamer outperforms existing methods by achieving more precise inter-object relationships and superior individual object modeling.

### A.13 Failure Cases and Limitations

While *DecompDreamer* excels at generating 3D assets composed of distinct object types, it still exhibits several limitations. Figures 20(a) and (b) illustrate the model's difficulty in decomposing a single entity into fine-grained subcomponents—such as segmenting a 3D human into semantically meaningful parts like the face, hands, or limbs—as well as in handling complex actions (e.g., *a bag with its zipper open*). In such intricate compositional scenarios, textual prompts often lack the granularity required to effectively supervise the generation process and can introduce ambiguity when lifting 2D semantics into 3D space. These cases may benefit from higher-order supervision signals, such as reference images or explicit spatial layouts.

### A.14 List of prompts

To ensure full transparency and reproducibility, this appendix provides the complete set of 30 compositional text prompts (Table 7) that comprise our evaluation benchmark. These prompts were used to generate the assets for both our automated quantitative metrics and our human preference studies.

Table 7: Full Evaluation Benchmark Prompts

| Prompt # | Prompt Text |
| --- | --- |
| 1 | A well-furnished bedroom with a Baroque-style bed with a red pillow on it, two wooden nightstands with pots on top of them, a square wooden table holding a decorative table lamp, a wooden wardrobe and a kid infront of the bed and a pet dog standing next to the bed between nightstand and wardrobe. |
| 2 | A well-furnished bedroom with a Baroque-style bed with a red pillow on it, two wooden nightstands, a square wooden table holding a decorative table lamp, a wooden wardrobe. |
| 3 | Three macaw parrots sharing a milkshake with three straws |
| 4 | A bedroom with a Baroque bed, two wooden nightstands, a square wooden table with a potting on it, and a wooden wardrobe |
| 5 | A knight in shining armor gallops on a brown horse, wearing a blue hat, holding a sword in his right hand. A princess sits beside him on the same horse, wearing a flowing gown and a jeweled crown |
| 6 | A bedroom with a bed, two wooden nightstands, a square wooden table with a table lamp on it, and a wooden wardrobe |
| 7 | A Wizard standing in front of a Wooden Desk, gazing into a Crystal Ball placed on the Wooden Desk, with a Stack of Ancient Spell Books sitting on the Wooden Desk and next to the crystal ball. |
| 8 | A knight in full shining armor galloping on a brown horse, wearing a blue hat, and holding a sword |
| 9 | Panda in a wizard hat sitting on a wooden chair and looking at a ficus in a pot |
| 10 | Two macaw parrots sharing a milkshake with two straws |
| 11 | A Teddy Bear pushing a Shopping Cart full of vegetables with right hand and holding a Handful Of Balloons in left hand |
| 12 | A Hero Character confronting a Dragon that protecting a Crystal, fantasy game scenario |
| 13 | A DSLR photo of a robot and dinosaur playing chess, high resolution |
| 14 | A macaw parrot drinking milshake with a straw |
| 15 | A hippo biting through a watermelon |
| 16 | A badger wearing a party hat and blowing out birthday candles on a cake |
| 17 | A baby bunny sitting on top of a stack of pancakes |
| 18 | A blue jay standing on a large basket of rainbow macarons |
| 19 | An astronaut riding a brown horse |
| 20 | A gummy bear playing the saxophone |
| 21 | A fox working on a jigsaw puzzle, cute, cartoon |
| 22 | A silver knight galloping a chestnut horse |
| 23 | A blue poison-dart frog sitting on a water lily, high detail 3d model. |
| 24 | A DSLR photo of a lion reading the newspaper |
| 25 | Michelangelo style statue of dog reading news on a cellphone |
| 26 | A zoomed out DSLR photo of a table with dim sum on it |
| 27 | A squirrel gesturing in front of an easel showing colorful pie charts |
| 28 | An artist is painting on a blank canvas |
| 29 | A piglet playing the piano |
| 30 | A DSLR photo of a humanoid robot playing the cello |

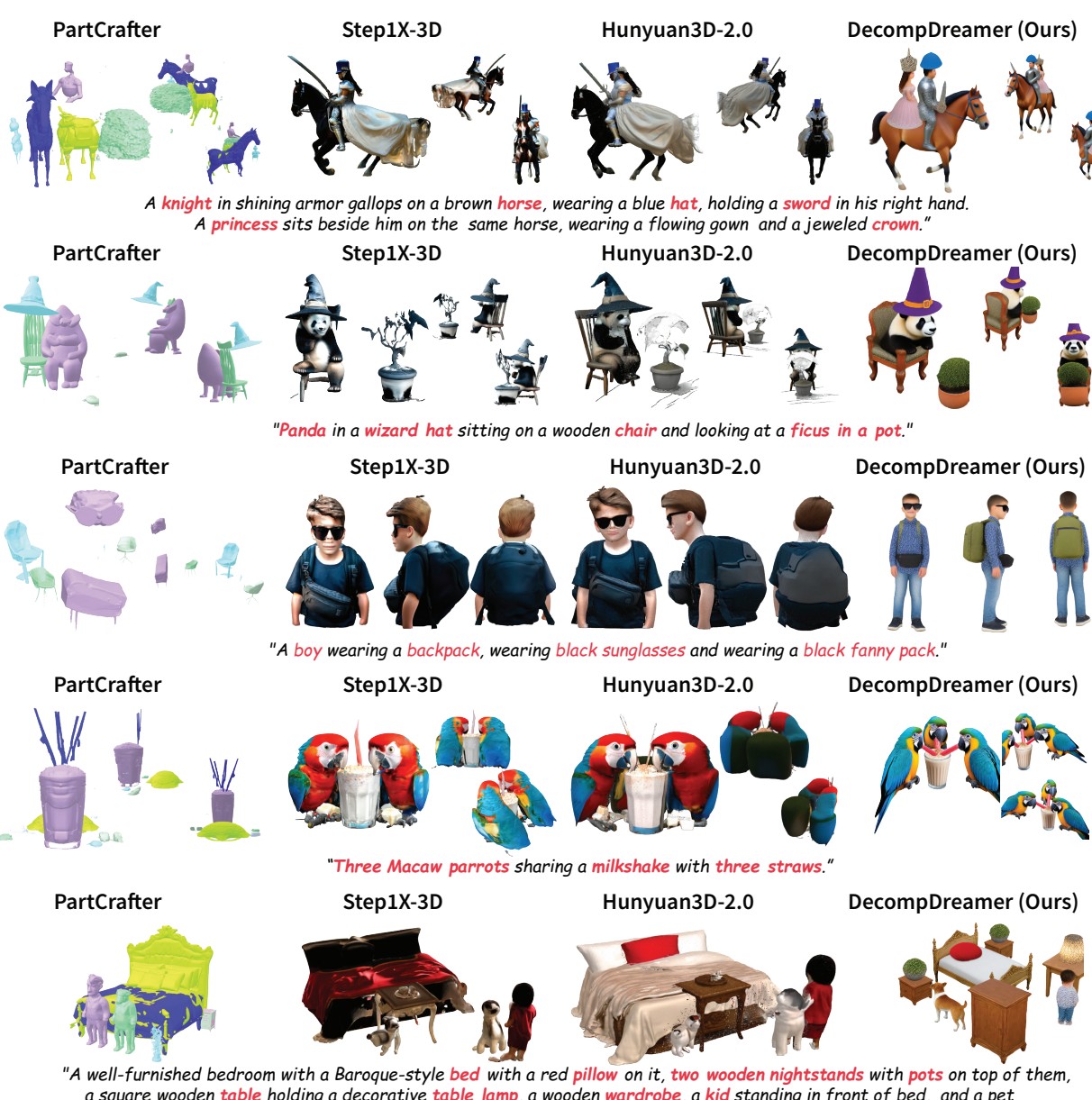

Figure 17: Qualitative comparison with between feed-forward 3D object generation methods such as PartCrafter, Step1X-3D, and Hunyaun3d-2.0, against DecompDreamer.

## A.15 Open-Source VLM Compatibility

To demonstrate our framework's compatibility with open-source models, we compared the structural initialization priors generated by GPT-4o against Qwen-VL. As shown in Table 8, the open-source VLM produces spatial coordinates, scales, and azimuths with minimal deviation from the proprietary baseline. Alongside these metrics, we also provide a qualitative result (Figure 19) demonstrating that the final 3D generated scene remains structurally accurate and visually consistent when using the Qwen-VL backbone. Because our spatial correction module is explicitly designed to absorb minor layout variations, this confirms the framework can be fully reproduced using open-source supervision.

Table 8: Relative Deviation of Initialization Priors (GPT-4o vs. Qwen-VL). Values show the overall vector percentage change (Euclidean deviation) from GPT-4o to Qwen-VL.

| Object | Δ Spatial Coords | Δ Angles | Δ Scale | Δ Azimuth |
|---|---|---|---|---|
| *Prompt 1: A knight in shining armor gallops on a brown horse...* | | | | |
| **1. Knight** | 5% | 0% | 8.3% | 0% |
| **2. Horse** | 5% | 0% | 11.1% | 0% |
| **3. Cap** | 5.4% | 0% | 0% | 0% |
| **4. Sword** | 5.12% | 11% | 10.0% | 0% |
| **5. Crown** | 5.1% | 0% | 0% | 0% |
| **6. Princess** | 7% | 0% | 8.3% | 0% |
| *Prompt 2: A Wizard standing in front of a Wooden Desk...* | | | | |
| **1. Wizard** | 11.2% | 3% | 8.5% | 0% |
| **2. Table** | 5.4% | 7% | 10% | 0% |
| **3. Crystal Ball** | 3.6% | 0% | 10.0% | 0% |
| **4. Book** | 3.7% | 0% | 0% | 0% |

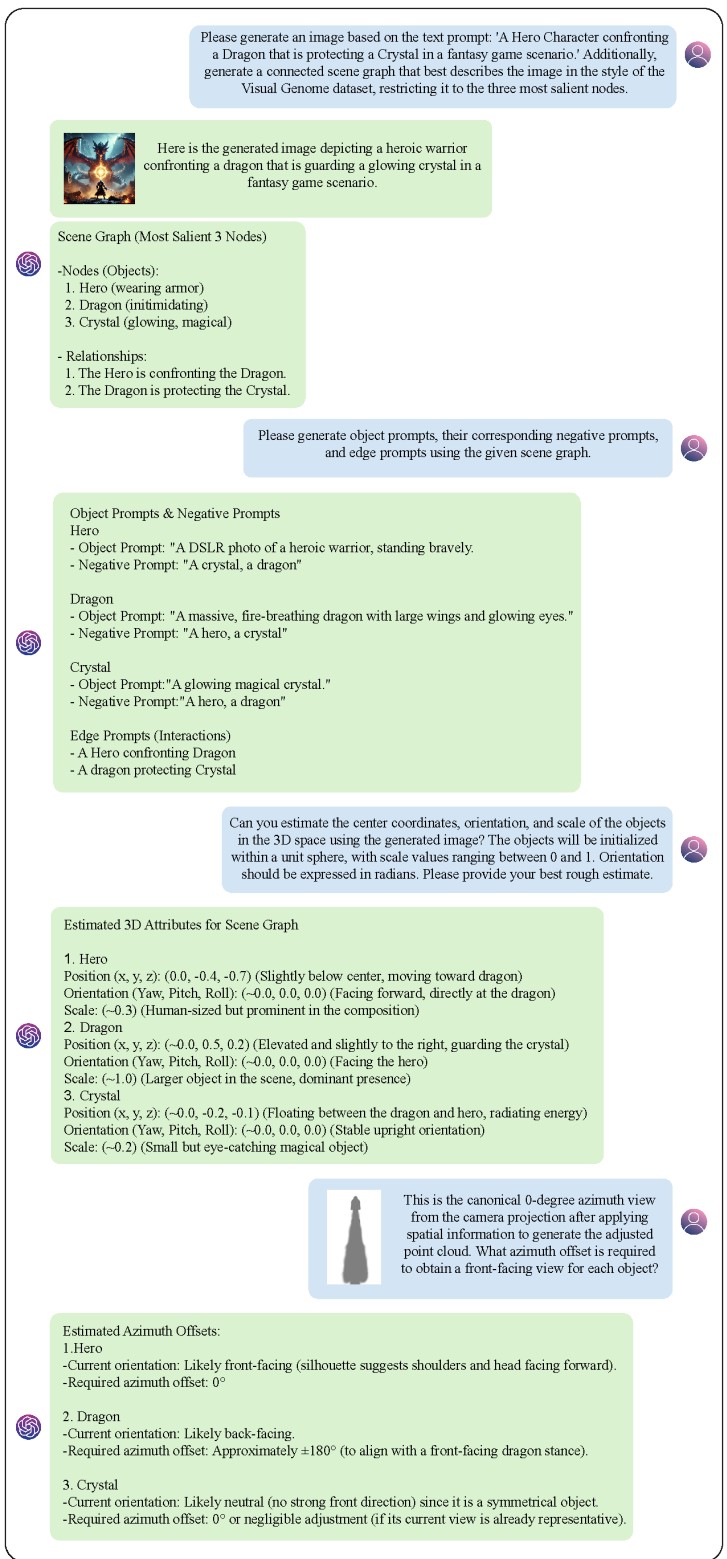

Figure 18: Example usage of ChatGPT in DecompDreamer for generating compositional 3D objects.

GPT-4o
Qwen-VL

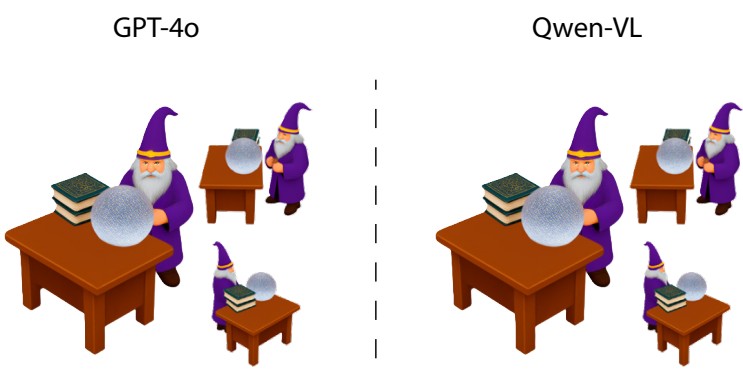

Figure 19: Qualitative comparison of the "Wizard" prompt utilizing proprietary GPT-4o (left) versus open-source Qwen-VL (right) initialization priors. Although the initial spatial estimations exhibit minor numerical deviations (as quantified in Table 8), ours spatial correction module successfully absorbs this variance. The final optimized geometry maintains strict relational fidelity, confirming the framework's robustness without reliance on proprietary VLM supervision.

(a)
(b)

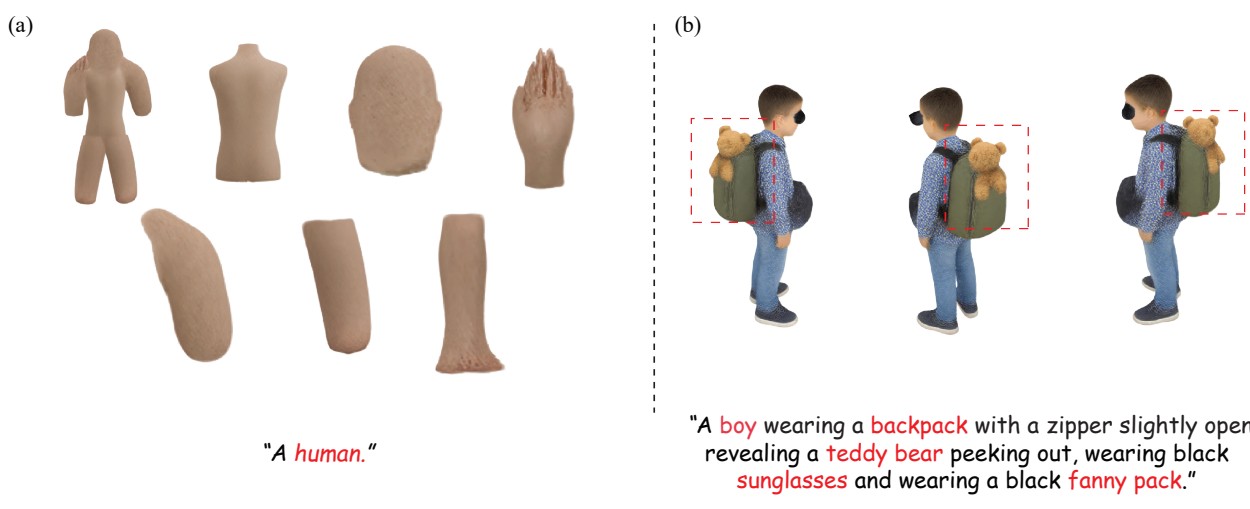

"A human."

"A boy wearing a backpack with a zipper slightly open revealing a teddy bear peeking out, wearing black sunglasses and wearing a black fanny pack."

Figure 20: Examples of failure cases. (a) Difficulty in decomposing a single entity into fine-grained subcomponents—such as segmenting human body parts. (b) A case where DecompDreamer fails to model complex actions, such as *a bag with its zipper open.*

Point-E Initialization
GALA3D
GraphDreamer
DecompDreamer (Ours)

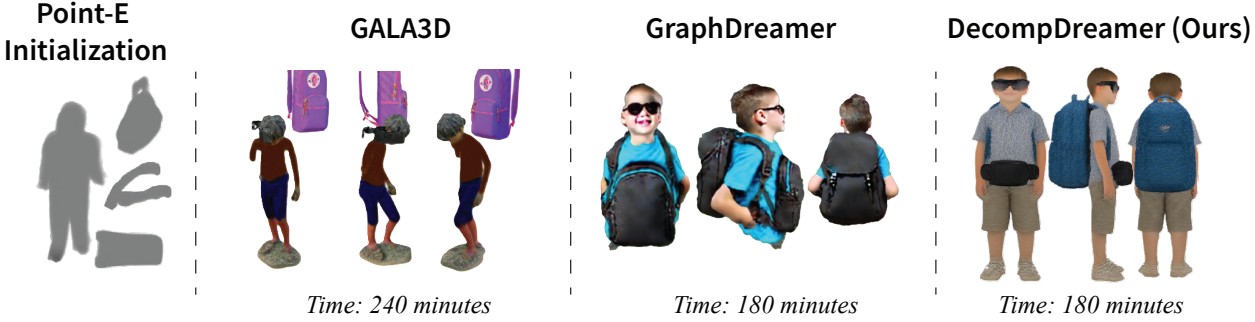

*Time: 240 minutes*
*Time: 180 minutes*
*Time: 180 minutes*

Figure 21: Qualitative comparison of GALA3D and GraphDreamer against DecompDreamer under the same Point-E initialization. Despite using the same low-fidelity coarse initializer, DecompDreamer produces more coherent object geometry and better preserves inter-object relationships in complex compositional prompts, while requiring comparable or lower runtime.

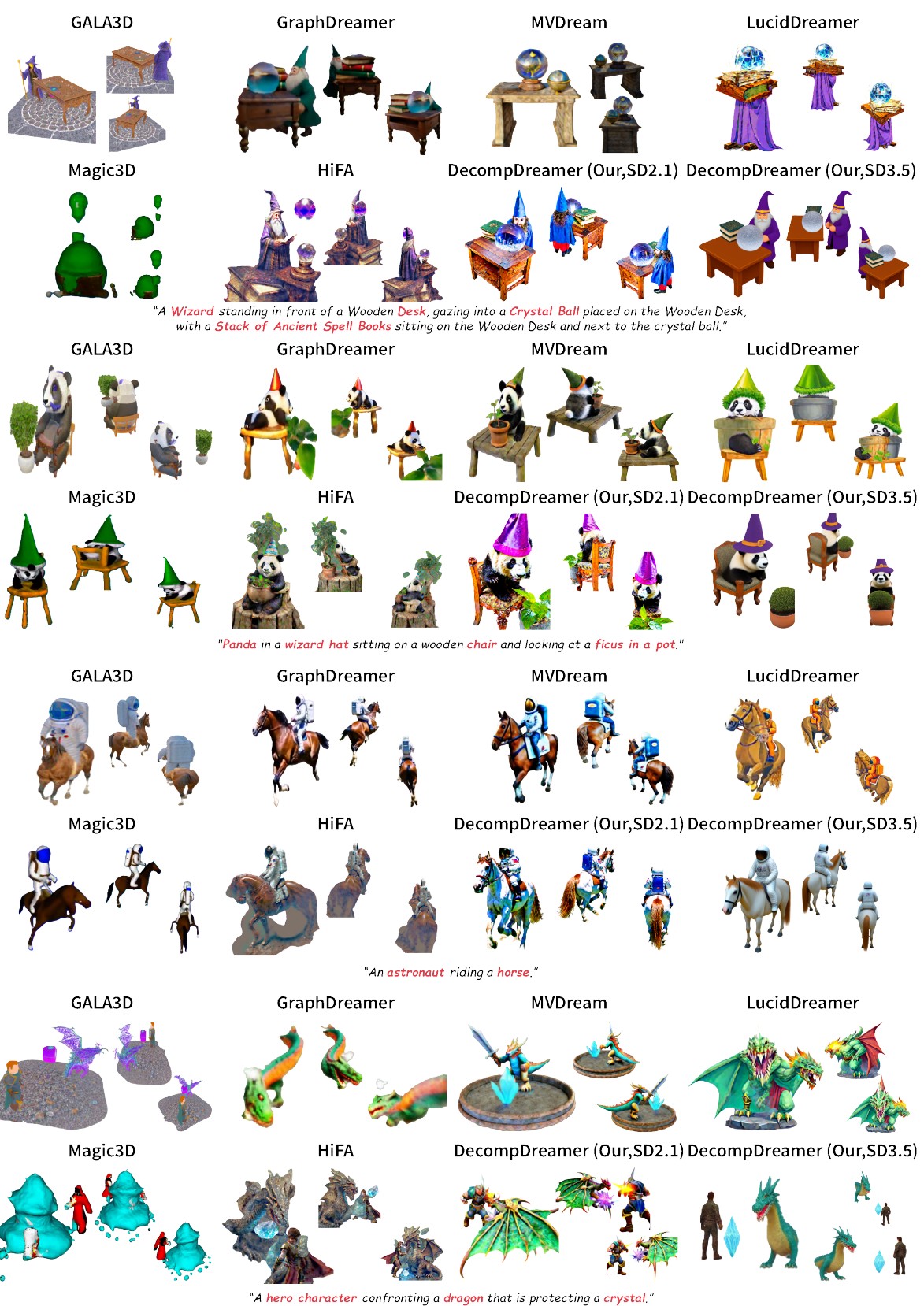

Figure 22: More qualitative comparison of GALA3D, GraphDreamer, MVDream, LucidDreamer, Magic3D and HiFA against DecompDreamer with SD2.1 and SD3.5 as backbones.

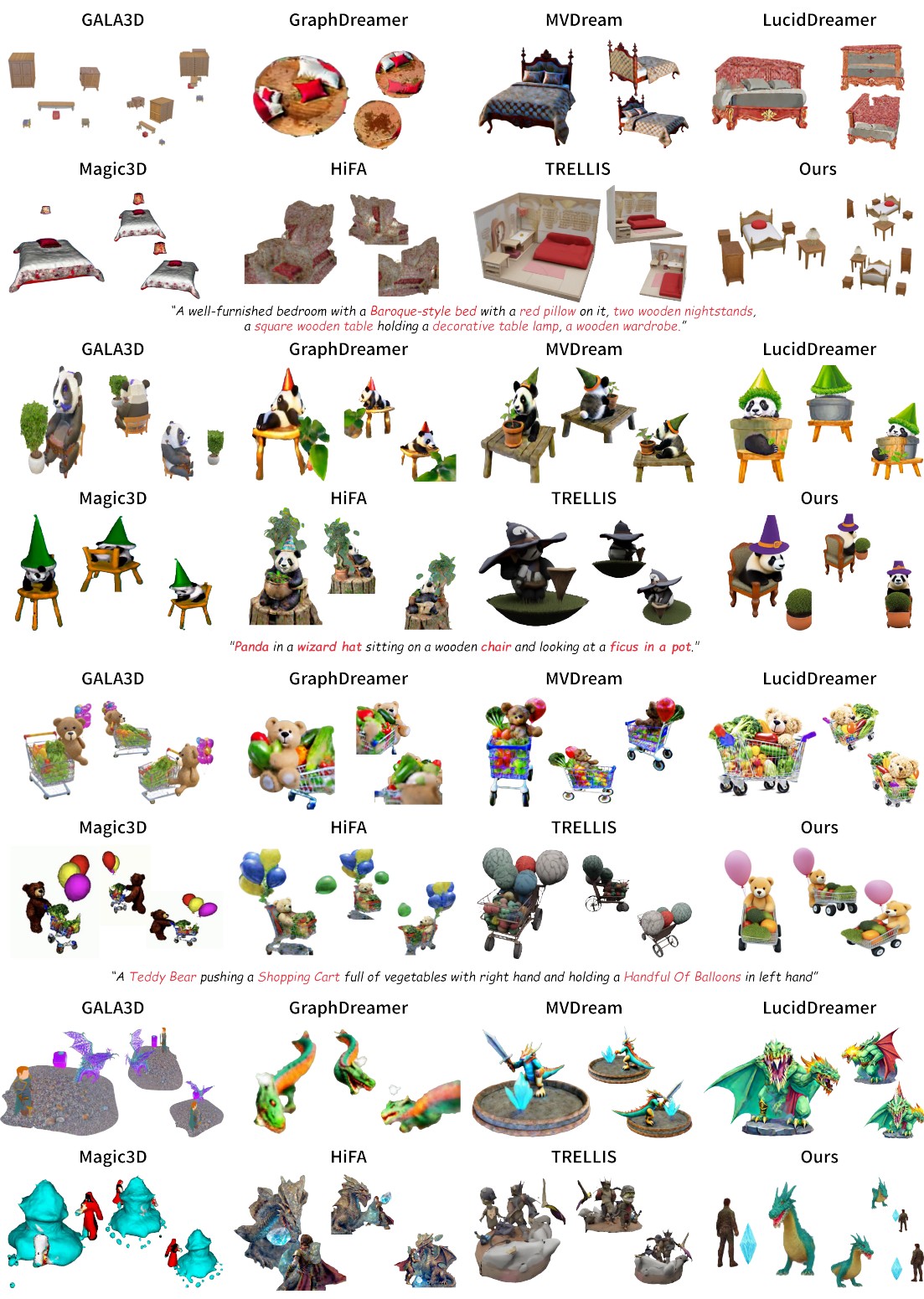

Figure 23: More qualitative comparison between the proposed DecompDreamer and state-of-the-art generators, including GALA3D Zhou et al. (2024), GraphDreamer Gao et al. (2024), Trellis Xiang et al. (2025), MVDream Shi et al. (2024), LucidDreamer Liang et al. (2024), Magic3D Lin et al. (2023) and HiFA Zhu et al. (2024).

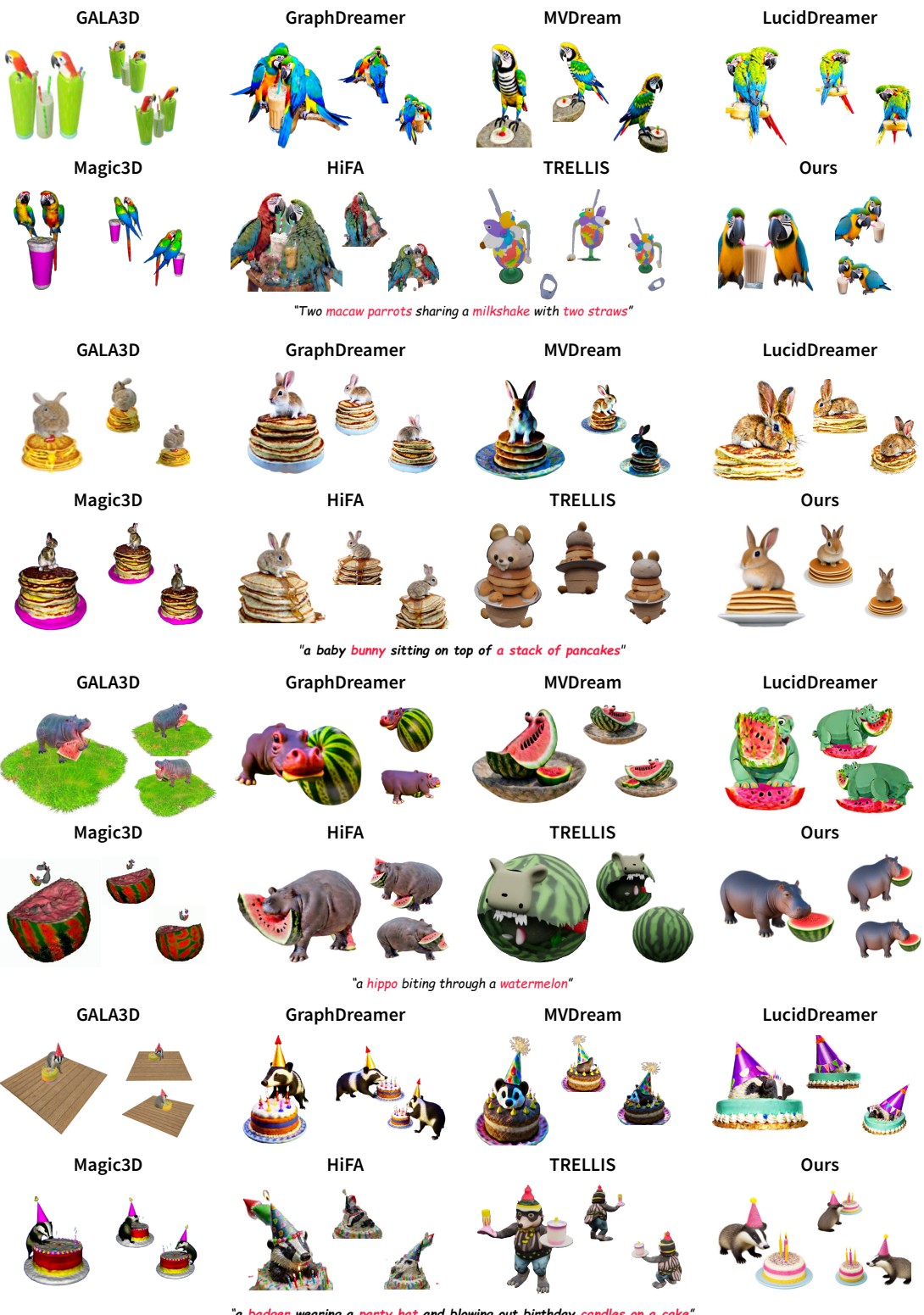

Figure 24: More qualitative comparison between the proposed DecompDreamer and state-of-the-art generators, including GALA3D Zhou et al. (2024), GraphDreamer Gao et al. (2024), Trellis Xiang et al. (2025), MVDream Shi et al. (2024), LucidDreamer Liang et al. (2024), Magic3D Lin et al. (2023) and HiFA Zhu et al. (2024).

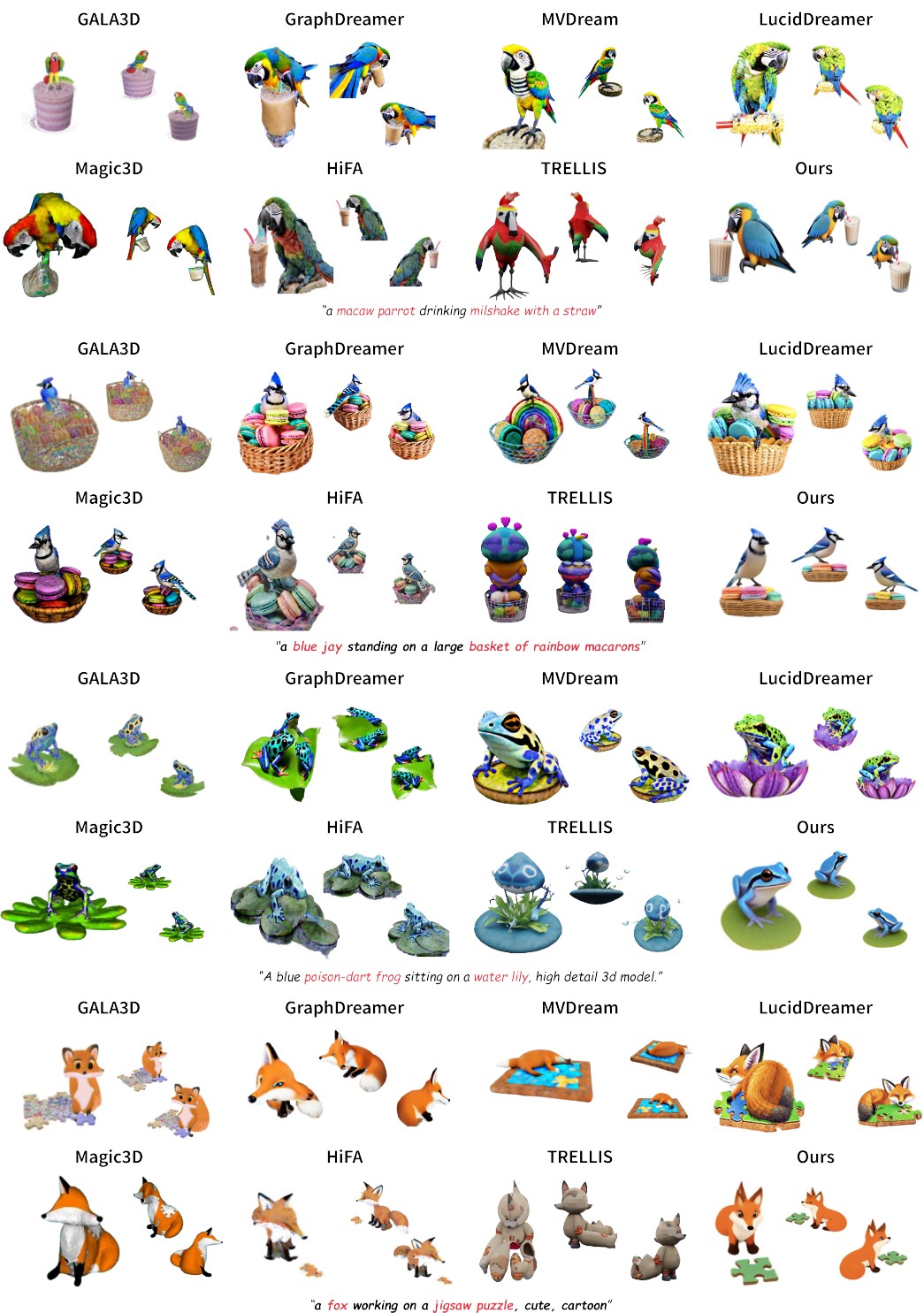

Figure 25: More qualitative comparison between the proposed DecompDreamer and state-of-the-art generators, including GALA3D Zhou et al. (2024), GraphDreamer Gao et al. (2024), Trellis Xiang et al. (2025), MVDream Shi et al. (2024), LucidDreamer Liang et al. (2024), Magic3D Lin et al. (2023) and HiFA Zhu et al. (2024).

