# OpenReview forum: "DecompDreamer: A Composition-Aware Curriculum for Structured 3D Asset Generation"
_TMLR — Decision pending for TMLR_

### Review · Reviewer_HC2m · 2026-03-31

**Summary Of Contributions:**

This paper introduces DecompDreamer, a new approach for compositional text-to-3D generation that focuses on improving how the model is optimized rather than how scenes are decomposed. The authors argue that existing methods struggle mainly because they try to learn everything at once, which leads to conflicting gradients and unstable results.

To solve this, DecompDreamer uses a staged, curriculum-style optimization process. It first learns the overall structure of a scene (capturing how objects relate to each other) ,inter-object relationships as stage 1, and only afterward focuses on refining the details of individual objects, individual object details as stage 2. By separating these steps over time, the method reduces interference between objectives and leads to more stable training.

This temporal decoupling reduces gradient conflicts and improves stability. The method is finally evaluated on complex multi-object prompts and shows strong improvements over prior work in terms of fidelity, coherence, and disentanglement.

**Additional Comments:**

see above

**Audience:**

Yes

**Audience Explanation:**

yes, see above

**Claims And Evidence:**

Yes

**Claims Explanation:**

The paper propose the idea of reframing compositional 3D generation as an optimization scheduling problem, which offers a fresh perspective on why existing methods struggle.The authors identify gradient conflicts as the core issue and propose a staged, curriculum-based solution that feels both intuitive and grounded in practical workflows. This insight is supported by theoretical foundation, where the problem is clearly framed as a combinatorial optimization challenge, and the role of conflicting gradients in multi-object generation is analyzed.
The paper also introduces an optimization strategies, ranging from holistic to staged. On the methodological side, the proposed two-stage approach is well designed, with additional elements like joint versus targeted optimization, view-aware supervision, and negative prompting .
 Empirically, the results are equally compelling, showing consistent improvements over few SOTA methods.

**Requested Changes:**

While the central idea of staged optimization is novel and compelling, much of the underlying methodology builds on existing components such as SDS, Gaussian splatting, and scene graphs. As a result, the main contribution appears to lie more in how these components are scheduled rather than in introducing fundamentally new modeling techniques, and clarifying this distinction would strengthen the paper.

 In addition, the framework relies heavily on a Vision-Language Model (ChatGPT-4o) for scene graph initialization, but the sensitivity of the method to noisy or incorrect outputs from this model is not sufficiently explored; further explanation and analysis would help readers better understand this dependency.

There are also some concerns regarding the evaluation, including the relatively small scale of the user studies, the potential bias introduced by GPT-based evaluation, and the limited details provided for reproducibility.

 Finally, the paper would benefit from a more explicit discussion of its limitations, particularly in terms of 1)failure cases, 2)scalability, and 3)real-world applicability, as additional clarification in these areas would make the work more complete.

---

> ### Author Response · Authors · 2026-04-30
>
> **Q1: While the central idea of staged optimization is novel and compelling...**
>
>  We thank the reviewer for their insightful observation. The reviewer is entirely correct that DecompDreamer builds upon established components like SDS and Gaussian Splatting. We consider this a core strength of our work rather than a limitation. By keeping the underlying components standard, we isolate and prove our central thesis: the fundamental bottleneck in compositional 3D generation is not a lack of new modeling techniques, but the optimization schedule itself. We have clarified this distinction in the introduction to explicitly frame our contribution as a solution to gradient conflict through temporal scheduling.
>
> Our primary conceptual contribution is the systematic taxonomy of prior heuristics detailed in Section 2.2. As we demonstrate, existing simultaneous and iterative methods predictably collapse under a combinatorial explosion of conflicting gradients. To mathematically quantify this phenomenon, we introduced the Phase Specific Divergence Rate in Section 6.1. As shown in Table 1, while prior methods diverge during refinement due to these conflicting gradients, our staged curriculum maintains stable, negative divergence rates.
>
> To empirically validate that the scheduling itself is the decisive factor, we refer to our ablation study in Section 6.3 and Table 6. The Baseline plus Staged Curriculum configuration, which utilizes only standard diffusion guidance without any auxiliary refinement modules, yields a $20 - 30$ % relative gain in relational fidelity and optimization stability over non staged baselines. This confirms that reframing the generation process as a temporal curriculum is a highly effective, standalone contribution that resolves the catastrophic divergence plaguing prior compositional frameworks.
>
> **Q2: In addition, the framework relies heavily on a Vision-Language Model...**
>
> To analyze how the framework handles incorrect spatial outputs from the VLM, we conducted a targeted stress test (detailed in Section 6.5 and Figure 9). We manually injected Gaussian noise (up to 0.2 units) into the predicted object centers. The framework successfully recovers from these errors via the Spatial Error Correction module (Section 5.2.3). By freezing object geometry and optimizing only the translation parameters using the joint edge loss, the diffusion model’s relational priors physically pull misaligned objects back into their correct interaction zones. Because the final alignment is governed by robust diffusion gradients rather than the raw VLM output, the framework is highly resilient to initial spatial imprecision.
>
> To mitigate incorrect scene graphs , we utilize a highly structured, multi-stage, image-anchored prompting strategy detailed in Appendix A.11. By forcing the VLM to extract the scene graph from a concrete 2D image blueprint rather than abstract zero-shot text reasoning, we significantly reduce the frequency of structural hallucinations.
>
> **Q3: There are also some concerns regarding the evaluation...**
>
> We address the scale of the user studies by computing 95% confidence intervals and conducting paired statistical tests ($p < 0.05$) to mathematically prove our method's robustness. Second, to eliminate potential evaluator bias for GPT-4o-based metrics, we conducted a human annotation study demonstrating a 91% agreement rate with our Relational Fidelity metric. We also successfully cross-validated our results using the Gemini-3.1-pro model. Finally, to ensure reproducibility, we have added the comprehensive list of our full 30-prompt evaluation benchmark in Appendix A.14 (Table 7).
>
> | Method | Rel. Fid. Human Agreement (%) | Rel. Fid. (GPT) | Rel. Fid. (Gemini) |
> | :--- | :---: | :---: | :---: |
> | Magic3D | 85 | 38.3 | 37.6 |
> | GraphDreamer | 89 | 53.5 | 53.5 |
> | CompGS | 91 | 72.3 | 70.5 |
> | GALA3D | 90 | 69.8 | 70.5 |
> | LucidDreamer | 87 | 32.4 | 32.3 |
> | MVDream | 86 | 47.3 | 47.3 |
> | Step1X-3D | 94 | 49.1 | 49.0 |
> | Hunyuan3D-2.0 | 95 | 48.1 | 48.6 |
> | Trellis | 93 | 48.4 | 48.5 |
> | **Ours** | **100** | **100.0** | **100.0** |
> | *Average* | *91%* | *-* | *-* |

---

> > ### Author Response · Authors · 2026-04-30
> >
> > **Q3: cont...**
> >
> >
> > | Method | GER ($\pm$ 95% CI) | GER ($p$-value) | User Study ($\pm$ 95% CI) | User Study ($p$-value) |
> > | :--- | :---: | :---: | :---: | :---: |
> > | Magic3D | 52.01 $\pm$ 19.37% | $p < 0.001$ | 0.0 $\pm$ 0.0% | $p < 0.001$ |
> > | GraphDreamer | 41.87 $\pm$ 19.64% | $p < 0.001$ | 9.0 $\pm$ 2.9% | $p < 0.001$ |
> > | CompGS | 20.90 $\pm$ 11.70% | $p = 0.005$ | 10.0 $\pm$ 3.1% | $p < 0.001$ |
> > | GALA3D | 26.39 $\pm$ 18.78% | $p < 0.05$ | 6.0 $\pm$ 2.6% | $p < 0.001$ |
> > | LucidDreamer | 72.22 $\pm$ 18.11% | $p < 0.001$ | 3.0 $\pm$ 1.8% | $p < 0.001$ |
> > | MVDream | 57.12 $\pm$ 17.74% | $p < 0.001$ | 0.0 $\pm$ 0.0% | $p < 0.001$ |
> > | Step1X-3D | 22.19 $\pm$ 11.03% | $p = 0.0068$ | 8.0 $\pm$ 2.8% | $p < 0.001$ |
> > | Hunyuan3D-2.0 | 22.45 $\pm$ 11.63% | $p < 0.001$ | 6.0 $\pm$ 2.4% | $p < 0.001$ |
> > | Trellis | 41.90 $\pm$ 16.52% | $p < 0.001$ | 6.0 $\pm$ 2.5% | $p < 0.001$ |
> > | **Ours** | **3.24 $\pm$ 4.41%** | - | **51.0 $\pm$ 5.1%** | - |
> >
> > **Q4: Finally, the paper would benefit from...**
> >
> > Regarding failure cases, as detailed in Appendix A.13 and Figure 20, DecompDreamer occasionally struggles with fine-grained intra-object decomposition, such as segmenting a single human mesh into distinct semantic parts like hands or faces. It also faces challenges modeling complex non-rigid object states, such as a backpack with a partially open zipper. These cases highlight the boundaries of current VLM spatial reasoning when dealing with tightly coupled geometries, presenting a clear avenue for future work.
> >
> > In terms of scalability, our framework scales linearly with the number of objects, taking approximately 12 minutes to complete a standard two-object scene. For our experiments, a single 40GB A100 GPU comfortably supports up to 11 distinct objects in a single scene, and increasing the VRAM would easily allow for more objects. However, to maximize real-world applicability, we believe that pairing fast feed-forward methods with our optimization regime will ultimately provide the most significant benefits for both generation time and scaling object count

---

### Review · Reviewer_5ekP · 2026-04-01

**Summary Of Contributions:**

The paper introduces DecompDreamer, a framework which generates 3D assets from text prompts. The authors argue that existing text-to-3D methods fail on complex scenes due to "gradient conflicts" arising from simultaneous optimization schedules. They fix this by proposing a two stage curriculum. In Stage 1, the model prioritizes inter-object relationships to establish a coarse structural layout. Then, they iteratively refine object details with view-aware, object-specific supervision. The pipeline is driven by a VLM like GPT-4o that generates a scene graph and spatial priors for coarse initialization. Additional technical contributions include a Spatial Error Correction module to fix misalignment, view-aware object supervision (azimuth correction) to fix orientation issues, and negative prompts to prevent feature blending between objects.

Key Strengths:

1. The "structure-then-detail" idea is well motivated and clear to understand, and explains how it tackles the problem of "gradient-conflicts" which arise from joint optimization of the scene objectives.
2. DecompDreamer achieves top performance across standard metrics and uniquely maintains stability as scene complexity scales. It achieves a perfect 100% Relational Fidelity and a 73.7% T3D Alignment score, significantly outperforming competitors like GraphDreamer and GALA3D on complex, multi-object prompts.
3. The paper presents a strong experimental evaluation, including qualitative comparisons, user studies, ablations, robustness tests, and runtime analysis.

Key Weaknesses:

1. Dependence on GPT-4o: Relying on GPT-4o for layouts and orientations creates reproducibility issues and introduces an unfair comparison, as the baselines do not benefit from this extra level of external supervision.
2. Relational fidelity is evaluated with GPT-4o, while the method also relies on GPT-4o-style scene understanding in the pipeline. This could bias the evaluation or make the results look stronger than they are. Having some human annotations and agreement rate analysis could strengthen the claims.
3. The paper still has limitations on finer-grained decomposition within a single object and more complex actions, which the authors acknowledge.

**Audience:**

Yes

**Audience Explanation:**

This work will be useful for audiences in the fields of 3D vision, text to 3D generation, and generative modelling in general.

**Claims And Evidence:**

Yes

**Claims Explanation:**

Yes the authors include a detailed description of their method, experiments, ablations and a diverse set of evaluations which show that DecompDreamer achieves much better performance than the baselines. The staged optimization design appears to improve compositional fidelity and stability on complex prompts.

**Requested Changes:**

I believe the following could strengthen the work:

1. An analysis of the failure modes of VLM (GPT-4o). Given that it is central to their method and is used for several stages of their pipeline generating the scene graph, producing the image blueprint, estimating object centers / sizes / orientations for coarse initialization, etc. Given that VLMs are prone to issues such as missed objects, wrong relationships, incorrect spatial attributes, orientation mistakes, and hallucinated structure. It would also help to discuss how robust the method is to these errors, what safeguards are used to reduce hallucinations, and which VLM prompting or decoding settings are used in practice, including hyperparameters such as temperature.
2. It would be useful to evaluate the pipeline with one or more open-source VLMs in addition to GPT-4o. Since the method depends on a VLM for scene graph generation, spatial attribute prediction, and azimuth estimation, showing that the approach also works with open models would make it easier to reproduce and cheaper to run. This would also help clarify whether the gains come mainly from the proposed optimization framework or from strong proprietary VLM supervision.
3. It will be useful if the relational fidelity evaluation can be corroborated by human evaluations and then agreement rate is measured compared to GPT-4o based evaluations. Since GPT-4o is also used in the pipeline for scene understanding and decomposition, relying on it again for evaluation may introduce bias or overstate performance.
4. Please provide the full prompt set and strengthen the reporting of the human evaluations. The appendix does describe the user-study setup and mentions randomized, double-blind evaluation, but the full 30-prompt benchmark is not provided. It will also be useful to provid e confidence interval tests, statistical significance tests etc, as the count of evaluation set is quite small.

---

> ### Author Response · Authors · 2026-04-30
>
> **Q1: An analysis of the failure modes of VLM (GPT-4o)...**
>
> We agree that a fundamentally flawed VLM output will degrade the downstream 3D result, and our framework does not claim to solve foundational VLM reasoning limits. However, the 2D image generation community has made significant strides in mitigating these exact layout and compositionality issues using LLMs as visual planners (e.g., LayoutGPT [1], LLM-grounded Diffusion [2]). Because we lift 2D information into 3D, our method inherits the benefits of these ongoing community advancements. Furthermore, our pipeline is highly modular: in the event of unusable VLM outputs, the coarse initial attributes (centers, sizes) can easily be defined or adjusted manually by a human user, and our 3D optimization pipeline will still successfully converge.
>
> Secondly, while we rely on the VLM for coarse initialization, the significant performance gains of DecompDreamer are derived strictly from our staged optimization curriculum. As shown in our ablation study (Section 6.3, Table 6), isolating the staged curriculum yields a 20% to 30% relative gain in relational fidelity over non-staged baselines, independent of VLM advantages.
> Thirdly, DecompDreamer does not blindly trust the spatial estimation from the VLM. To quantify our robustness to VLM spatial errors, we conducted a stress test by adding Gaussian noise (up to 0.2 normalized units) to predicted object centers (Appendix A.10, Figure 9). Our spatial correction module successfully mitigated these errors. By freezing object geometry and optimizing only the translation parameters using the joint edge loss, the diffusion priors physically pull misaligned objects into their correct interaction zones. Because actual physical alignment is governed by diffusion gradients rather than raw VLM outputs, the framework functions accurately even with noisy initializations.
>
> Finally, to explicitly address the reviewer’s question regarding safeguards against VLM hallucinations, we employ a highly structured, multi-stage, image-anchored prompting pipeline (detailed in Appendix A.11 and illustrated in Figure 18). Rather than relying on zero-shot text reasoning, our pipeline forces the VLM to ground all structural estimates in a concrete 2D visual blueprint. As shown in the figure, the VLM first generates a 2D image to extract a visually verified scene graph. It then systematically decomposes this graph into isolated positive and negative prompts for each object, estimates bounded 3D coordinates, and finally performs a canonical orientation alignment using 2D silhouette projections to correct facing directions.In practice, we execute all GPT-4o queries with a temperature of 0.0 to ensure maximum deterministic rigidity and reproducibility. This multi-modal approach effectively neutralizes the most common VLM failure modes before 3D generation even begins.
>
> **Q2: It would be useful to evaluate the pipeline with one or more open-source...**
>
> First, as shown in Section A.8 and Table 6, the significant performance gains of DecompDreamer are derived strictly from the staged optimization curriculum. The Baseline + Staged Curriculum configuration achieves large improvements over all non-staged baselines without relying on newer guidance techniques. This confirms that temporal decoupling is the primary driver of our results, operating independently of the VLM backbone.
>
> Second, to directly evaluate open-source compatibility, we compared the initialization priors from GPT-4o against the latest open-source models, such as Qwen-VL. We have added a qualitative comparison of these open-source initialization priors to Appendix A.15 (Table  8 and Figure 19). Because our pipeline utilizes TripoSR for the actual 3D object initialization based on text prompts, the primary variable introduced by switching VLMs is merely a minor variance in the initial 3D layout.
>
> As demonstrated in our Spatial Error Correction stress tests in Figure 9, DecompDreamer is explicitly designed to absorb these minor coordinate shifts. The spatial correction module easily fixes misalignments by physically pulling objects into the correct interaction zones before fine-grained refinement begins. Therefore, because open-source VLMs provide sufficiently accurate starting priors, and our spatial correction module neutralizes any residual layout noise, the final 3D generation quality remains consistently high without any reliance on proprietary VLM supervision.

---

> ### Author Response · Authors · 2026-04-30
>
> **Q2: cont..**
>
> **Prompt 1: "A knight in shining armor gallops on a brown horse..."**
>
> | Object | Δ Spatial Coords | Δ Angles | Δ Scale | Δ Azimuth |
> | :--- | :--- | :--- | :--- | :--- |
> | **1. Knight** | `5%` | `0%` | `8.3%` | `0%` |
> | **2. Horse** | `5%` | `0%` | `11.1%` | `0%` |
> | **3. Cap** | `5.4%` | `0%` | `0%` | `0%` |
> | **4. Sword** | `5.12%` | `11%` | `10.0%` | `0%` |
> | **5. Crown** | `5.1%` | `0%` | `0%` | `0%` |
> | **6. Princess** | `7%` | `0%` | `8.3%` | `0%` |
>
> **Prompt 2: "A Wizard standing in front of a Wooden Desk..."**
>
> | Object | Δ Spatial Coords | Δ Angles | Δ Scale | Δ Azimuth |
> | :--- | :--- | :--- | :--- | :--- |
> | **1. Wizard** | `11.2%` | `3%` | `8.5%` | `0%` |
> | **2. Table** | `5.4%` | `7%` | `10.0%` | `0%` |
> | **3. Crystal Ball** | `3.6%` | `0%` | `10.0%` | `0%` |
> | **4. Book** | `3.7%` | `0%` | `0%` | `0%` |
>
> **Q3: It will be useful if the relational fidelity evaluation can be corroborated**
>
> To explicitly validate our automated metric, we conducted a human annotation study where 5 independent human evaluators answered the exact same relational VQA prompts used by the GPT-4o evaluator across our 15 prompts. We observed a high raw agreement rate of 91% between human judgments and GPT-4o. This confirms that the VLM serves as a faithful, unbiased proxy for human perception of compositional correctness.
>
> Furthermore, to ensure the relational fidelity score is entirely independent of any GPT-specific artifacts or stylistic preferences, we re-evaluated our benchmark using Gemini 3.1 pro. The relative performance rankings remained identical (within $\pm 2$ %), with DecompDreamer consistently outperforming all baselines. This cross-model validation proves the robustness of our framework's gains and confirms they are not the result of evaluator bias.
>
> | Method | Rel. Fid. Human Agreement (%) | Rel. Fid. (GPT) | Rel. Fid. (Gemini) |
> | :--- | :---: | :---: | :---: |
> | Magic3D | 85 | 38.3 | 37.6 |
> | GraphDreamer | 89 | 53.5 | 53.5 |
> | CompGS | 91 | 72.3 | 70.5 |
> | GALA3D | 90 | 69.8 | 70.5 |
> | LucidDreamer | 87 | 32.4 | 32.3 |
> | MVDream | 86 | 47.3 | 47.3 |
> | Step1X-3D | 94 | 49.1 | 49.0 |
> | Hunyuan3D-2.0 | 95 | 48.1 | 48.6 |
> | Trellis | 93 | 48.4 | 48.5 |
> | **Ours** | **100** | **100.0** | **100.0** |
> | *Average* | *91%* | *-* | *-* |
>
> **Q4: Please provide the full prompt set and strengthen the reporting of the human evaluations...**
>
> We utilize the same 30 compositional prompts for our quantitative evaluation as those featured in our qualitative results. We have now compiled this full 30-prompt benchmark into a single comprehensive list in Appendix A.14 (Table 7).
>
> To strengthen the statistical rigor of our human evaluations, we now report 95% confidence intervals for all human-derived metrics. Additionally, we conducted paired statistical significance tests (a paired t-test for GER and an exact binomial test for user preference) to compare DecompDreamer against the baselines. The resulting p-values confirm that the improvements achieved by our method are statistically significant (p < 0.05).  By incorporating these confidence intervals and significance tests, we provide rigorous evidence that our framework's superiority is robust and not an artifact of the sample size.
>
> | Method | GER ($\pm$ 95% CI) | GER ($p$-value) | User Study ($\pm$ 95% CI) | User Study ($p$-value) |
> | :--- | :---: | :---: | :---: | :---: |
> | Magic3D | 52.01 $\pm$ 19.37% | $p < 0.001$ | 0.0 $\pm$ 0.0% | $p < 0.001$ |
> | GraphDreamer | 41.87 $\pm$ 19.64% | $p < 0.001$ | 9.0 $\pm$ 2.9% | $p < 0.001$ |
> | CompGS | 20.90 $\pm$ 11.70% | $p = 0.005$ | 10.0 $\pm$ 3.1% | $p < 0.001$ |
> | GALA3D | 26.39 $\pm$ 18.78% | $p < 0.05$ | 6.0 $\pm$ 2.6% | $p < 0.001$ |
> | LucidDreamer | 72.22 $\pm$ 18.11% | $p < 0.001$ | 3.0 $\pm$ 1.8% | $p < 0.001$ |
> | MVDream | 57.12 $\pm$ 17.74% | $p < 0.001$ | 0.0 $\pm$ 0.0% | $p < 0.001$ |
> | Step1X-3D | 22.19 $\pm$ 11.03% | $p = 0.0068$ | 8.0 $\pm$ 2.8% | $p < 0.001$ |
> | Hunyuan3D-2.0 | 22.45 $\pm$ 11.63% | $p < 0.001$ | 6.0 $\pm$ 2.4% | $p < 0.001$ |
> | Trellis | 41.90 $\pm$ 16.52% | $p < 0.001$ | 6.0 $\pm$ 2.5% | $p < 0.001$ |
> | **Ours** | **3.24 $\pm$ 4.41%** | - | **51.0 $\pm$ 5.1%** | - |
>
> **References**
>
> [1] Feng, Weixi, et al. "Layoutgpt: Compositional visual planning and generation with large language models." Advances in Neural Information Processing Systems 36 (2023): 18225-18250.
>
> [2] Lian, Long, et al. "Llm-grounded diffusion: Enhancing prompt understanding of text-to-image diffusion models with large language models." arXiv preprint arXiv:2305.13655 (2023).

---

### Review · Reviewer_3LJX · 2026-05-06

**Summary Of Contributions:**

This paper address the problem of compositional Text-to-3D generation problem. It argues/its hypothesis is that failure is fundamental to their optimization schedules and reframing this genetation as a optimization shceuling problem.

This paper proposes DecompDreamer as a novel staged optimization strategy transforming the intractable problem of joint compositional generation into a sequence of manageable sub-problem.

This paper aims in handling the text-to-3D generation problem in a new direction which is interesting and novel.

**Audience:**

Yes

**Audience Explanation:**

Intersting paper reformulating the training framework of text-to-3D generation.

**Broader Impact Concerns:**

yes.

This paper reframe a key-question in text-to-3D generation problem.

**Claims And Evidence:**

Yes

**Claims Explanation:**

Yes. There are extensive experiments showcasting the evidence.

**Requested Changes:**

This paper aims in arguing thate this failure from text-to-3D compositional generation is fundamental to their optimization schedules, as simultaneous or iterative heuristics predictably collapse under a combinatorial explosion. However, I would like to see more evidence supporting this hypothesis.

In table2, there is only  6 objects evaluted, making it less convincing. Is there more benchmarks to be tested?

---

> ### Author Response · Authors · 2026-05-08
>
> We sincerely thank the reviewer for recognizing the novelty of our framework and the value of reframing text-to-3D generation as an optimization scheduling problem. We appreciate the opportunity to clarify the scope of our evaluations and highlight the specific evidence supporting our hypothesis.
>
> **Q: I would like to see more evidence...**
>
> We provide direct, empirical evidence for this optimization collapse in Section 6.1:
> * **Visual Evidence:** In Figures 5 and 10, we plot the actual loss trajectories. For complex prompts, the iterative heuristic (GraphDreamer) shows rapid divergence, and the object and edge losses explode upwards as the gradients conflict. In contrast, DecompDreamer's staged curriculum shows a clean, stable descent.
> * **Quantitative Evidence:** In Table 1, we introduce a Phase-Specific Divergence Rate ($D_{rate}$) metric to quantify optimization stability and validate the theoretical analysis of gradient conflicts. During the refinement phase, baselines exhibit positive slopes ($D_{rate}>0$), quantitatively confirming optimization divergence. Our method maintains stability, exhibiting negative slopes ($D_{rate}<0$) throughout the entire generation process.
>
> **Q: In table 2, there is...**
>
> It appears there may be a slight misunderstanding caused by the phrasing in the caption of Table 2. The phrase "for generating 6 objects" in that caption refers only to the Execution Time column (as a standardized benchmark for speed). Our actual quantitative evaluation (CLIP, Pick-a-Pic, Text-to-3D Alignment, etc.) was conducted over a comprehensive benchmark of 30 distinct compositional prompts, varying in complexity from 2 up to 11 objects. To prevent future confusion, we have revised the caption of Table 2 in our manuscript to make this distinction clear.
>
> **Q: Is there more benchmarks...**
>
> To further address the request for broader benchmarks, Appendix A.6 (Table 5) breaks down our 30-prompt evaluation (the full list of prompts can be found in Appendix A.15, Table 7) to explicitly compare performance on simpler scenes ($\le3$ objects) versus highly complex scenes ($>3$ objects).
>
> As detailed in Table 5 of the Appendix, the baselines perform adequately on simple prompts but suffer catastrophic performance drops when combinatorial complexity increases (e.g., GraphDreamer's alignment drops from 50.0% to 16.5%). In contrast, DecompDreamer maintains state-of-the-art stability and semantic alignment regardless of scene complexity.
>
> We hope this clarifies that our framework has been rigorously evaluated across a broad spectrum of compositional complexities, and that the requested evidence for the optimization collapse is fundamentally proven in Section 6.1.

---

> > ### Comment · Reviewer_3LJX · 2026-05-20
> > **question about the claims**
> >
> > The observation on the claims are neat and interesting. I would like to further discuss about it.
> > 1. In the paper, the failure modes of holistic/simultaneous optimization are mainly attributed to gradient conflicts between relational and object-level objectives.
> >
> > I’m curious whether you explored any more direct measurements of this phenomenon, such as gradient cosine similarity, gradient norm interference, or task conflict analysis across optimization stages?
> >
> > 2. In this paper, it proposes that we should used staged optimization, which is neat. However, I’m wondering whether the gains come specifically from temporal decoupling itself, or more generally from optimization regularization / stabilization effects?
> >
> > 3. The paper links divergence in iterative optimization to combinatorial optimization conflicts. Did you run any controlled experiments varying object count or relational complexity independently to isolate this effect?

---

> > > ### Author Response · Authors · 2026-05-20
> > >
> > > We appreciate your thoughtful follow up questions.
> > >
> > > **Q: I’m curious whether you explored any more direct measurements of this phenomenon....**
> > >
> > > In our current manuscript, we utilize the Phase-Specific Divergence Rate ($D_{rate}$) and loss trajectory analysis as macroscopic, empirical proxies for gradient conflict. We did not compute explicit gradient cosine similarity or norm interference during the optimization steps, primarily due to the immense computational overhead of tracking high-dimensional SDS gradients across thousands of Gaussians. Performing separate backward passes to isolate task-specific gradients at every iteration is prohibitively expensive in a 3DGS-SDS pipeline. However, the chaotic tug of war we observe in the loss curves of iterative methods is the direct downstream manifestation of low or negative cosine similarity between relational and object-level gradients.
> > >
> > > **Q: However, I’m wondering whether the gains come specifically from temporal decoupling itself....**
> > >
> > > We specifically designed an ablation study to answer this exact question, which can be found in Table 6 of Appendix A.8. In this table, the Baseline + Staged Curriculum variant isolates the temporal scheduling mechanism entirely, removing auxiliary regularizations such as negative prompts, view-aware alignment, and advanced flow guidance. This barebones staged version still achieves substantial gains over the iterative and simultaneous baselines. This confirms that while our regularizations improve local fidelity and disentanglement, the fundamental stability and prevention of catastrophic collapse come specifically from the temporal decoupling of the objectives.
> > >
> > > **Q: Did you run any controlled experiments varying object count or relational complexity independently to isolate this effect?**
> > >
> > > We did conduct controlled evaluations isolating this effect, which are detailed in Table 5 of the appendix. We stratified our benchmark into scenes with low relational complexity (3 or fewer objects) and high relational complexity (more than 3 objects). As shown in Table 5, iterative methods like GraphDreamer perform adequately on simpler scenes but experience a catastrophic drop in Text-to-3D Alignment (from 50.0% to 16.5%) when forced to handle the combinatorial explosion of higher object counts. Our method maintains high alignment scores regardless of the object count, directly linking the failure of prior iterative optimization to combinatorial scaling limits.
> > >
> > > We hope these clarifications address your questions.

---

### Decision · Action_Editor_F5mr · 2026-06-18

**Recommendation:** Accept as is

**Additional Comments:**

The recommendation to accept this paper is supported by all reviewers. Comments include:

"The paper goes beyond being sound and relevant by offering a clear and useful reframing of compositional text-to-3D generation as an optimization-scheduling problem. The staged “structure-then-detail” curriculum is ... well motivated, and supported by strong quantitative results, ablations, and qualitative evidence. While the reliance on GPT-4o remains a limitation, the author response reasonably addresses this through robustness tests, open-source VLM analysis, and human/Gemini validation of the evaluation."

"I am satisfied with their clarifications and additional analyses. They clearly explain that the main contribution is the optimization schedule rather than new modeling components, which I find reasonable and convincing. They also address my concerns about the reliance on the Vision-Language Model by adding a stress test with noisy object centers and explaining how the Spatial Error Correction module improves robustness. The additional evaluation details, including confidence intervals, statistical tests, human agreement, and cross-validation with another model, make the results more convincing. The authors also added a clearer discussion of limitations, including failure cases, scalability, and real-world applicability. Overall, my concerns have been adequately addressed."

The authors are strongly advised to consider the specific suggestions offered by the reviewers in preparing their final manuscript, and to incorporate all aspects of the discussion within the final version.

**Audience:**

Yes

**Audience Explanation:**

It is of general interest to researchers interested in 3D generation.

**Claims And Evidence:**

Yes

**Claims Explanation:**

This paper frames compositional text-to-3D generation as an optimization scheduling problem. The approach uses a staged curriculum that first develops the scene structure, then object relationships, and finally fills in the detail, which is both intuitive and well motivated from a technical perspective. It builds on a variety of established components (e.g., Gaussian splatting, scene graphs, etc.), with innovation in how these are scheduled. The empirical evaluation is thorough, with strong consistent gains over prior methods, and results that are well-supported by ablations and analysis of the stability across scene/prompt complexity. In particular, the authors were able to isolate the contributions of temporal decoupling of the competing objectives from auxiliary components, showing that the scheduling strategy itself is the primary driver of the improved performance. Although the approach's pipeline is dependent on external components (e.g., GPT-4o), which adds a proprietary barrier and may slightly limit reproducibility, overall the work offers a practical and effective method for compositional 3D generation.